# Design and results of the ice sheet model initialisation experiments initMIP-Greenland: an ISMIP6 intercomparison

Heiko Goelzer 1,2, Sophie Nowicki 3, Tamsin Edwards 4, Matthew Beckley 3, Ayako Abe-Ouchi 5, Andy Aschwanden 6, Reinhard Calov 7, Olivier Gagliardini 8, Fabien Gillet-Chaulet 8 , Nicholas R. Golledge 9, Jonathan Gregory 10,11, Ralf Greve 12, Angelika Humbert 13,14, Philippe Huybrechts 15, Joseph H. Kennedy 16,17 , Eric Larour 18, William H. Lipscomb 19,20, Sébastien Le clec´h 21, Victoria Lee 22, Mathieu Morlighem 23, Frank Pattyn 2, Antony J. Payne 22, Christian Rodehacke 24,13, Martin Rückamp 13, Fuyuki Saito 25, Nicole Schlegel 18, Helene Seroussi 18, Andrew Shepherd 26, Sainan Sun 2, Roderik van de Wal 1, Florian A. Ziemen 27

(1) Utrecht University, Institute for Marine and Atmospheric Research (IMAU), Utrecht, Netherlands
(2) Laboratoire de Glaciologie, Université Libre de Bruxelles, Brussels, Belgium
(3) NASA GSFC, Cryospheric Sciences Branch, Greenbelt, USA
(4) School of Environment, Earth & Ecosystem Sciences, The Open University, Milton Keynes, United Kingdom
(5) Atmosphere Ocean Research Institute, University of Tokyo, Kashiwa, Japan
(6) Geophysical Institute, University of Alaska Fairbanks, USA
(7) Potsdam Institute for Climate Impact Research, Potsdam, Germany
(8) Univ. Grenoble Alpes, CNRS, IRD, Grenoble INP, IGE, F-38000 Grenoble, France
(9) Antarctic Research Centre, Victoria University of Wellington, Wellington, New Zealand
(10) Department of Meteorology, University of Reading, Reading, United Kingdom
(11) Met Office Hadley Center, Exeter, United Kingdom
(12) Institute of Low Temperature Science, Hokkaido University, Sapporo, Japan
(13) Alfred Wegener Institute for Polar and Marine Research, Bremerhaven, Germany
(14) University of Bremen, Bremen, Germany
(15) Vrije Universiteit Brussel, Brussels, Belgium
(16) Climate Change Science Institute, Oak Ridge National Laboratory, Oak Ridge, USA
(17) Computational Sciences and Engineering Division, Oak Ridge National Laboratory, Oak Ridge, USA
(18) Jet Propulsion Laboratory, California Institute of Technology, Pasadena, USA
(19) Los Alamos National Laboratory, Los Alamos, USA
(20) National Center for Atmospheric Research, Boulder, USA
(21) LSCE/IPSL, Laboratoire des Sciences du Climat et de l'Environnement, CEA-CNRS-UVSQ, Gif-sur-Yvette, France
(22) University of Bristol, Bristol, United Kingdom
(23) University of California Irvine, Irvine, USA
(24) Danish Meteorological Institute, Copenhagen, Denmark
(25) Japan Agency for Marine-Earth Science and Technology, Yokohama, Japan
(26) School of Earth and Environment, University of Leeds, United Kingdom
(27) Max Planck Institute for Meteorology, Hamburg, Germany

*Correspondence to*: Heiko Goelzer (h.goelzer@uu.nl)

**Abstract.**

Earlier large-scale Greenland ice sheet sea-level projections (e.g., those run during the ice2sea and SeaRISE initiatives) have shown that ice sheet initial conditions have a large effect on the projections and give rise to important uncertainties. The goal of the initMIP-Greenland intercomparison exercise is to compare, evaluate and improve the initialisation techniques used in the ice sheet modelling community and to estimate the associated uncertainties in modelled mass changes. initMIP-Greenland is the first in a series of ice sheet model intercomparison activities within ISMIP6 (the Ice Sheet Model Intercomparison Project for CMIP6), which is the primary activity within the Coupled Model Intercomparison Project – phase 6 (CMIP6) focusing on the ice sheets. Two experiments for the large-scale Greenland ice sheet have been designed to allow intercomparison between participating models of 1) the initial present-day state of the ice sheet and 2) the response in two idealised forward experiments. The forward experiments serve to evaluate the initialisation in terms of model drift (forward run without additional forcing) and in response to a large perturbation (prescribed surface mass balance anomaly), and should not be interpreted as sea-level projections. We present and discuss results that highlight the diversity of data sets, boundary conditions and initialisation techniques used in the community to generate initial states of the Greenland ice sheet. We find good agreement across the ensemble for the dynamic response to surface mass balance changes in areas where the simulated ice sheets overlap, but differences arising from the initial size of the ice sheet. The model drift in the control experiment is reduced for models that participated in earlier intercomparison exercises.

## 1   Introduction

Ice sheet model intercomparison exercises have a long history, going back to the advent of large-scale ice sheet models in the early 1990s. The first intercomparison project (EISMINT, the European Ice Sheet Modelling INiTiative; Huybrechts et al., 1996) defined three levels of possible comparisons that could be distinguished. EISMINT and later following comparisons include (1) schematic experiments with identical model setup and boundary conditions between models (e.g., Huybrechts et al., 1996; Pattyn et al., 2008; Pattyn et al., 2012; Pattyn et al., 2013), (2) experiments allowing individual modelling decisions (e.g., Payne et al., 2000; Calov et al., 2010; Asay-Davis et al., 2016), and (3) experiments of models applied to real ice sheets (e.g., Shannon et al., 2012; Edwards et al., 2014b; Bindschadler et al., 2013, Nowicki et al., 2013a, b). In this genealogy, the present intercomparison is a type 3 experiment with ice sheets models applied to simulate the large-scale present-day Greenland ice sheet. The role of this study is to assess the impact of initialisation on model behaviour; it is a precursor to ice-sheet mass budget projections made using climate forcing for the atmosphere and ocean. The initMIP-Greenland project is the first intercomparison within ISMIP6, the Ice Sheet Model Intercomparison Project for CMIP6 (Nowicki et al., 2016), which is the primary activity within the Coupled Model Intercomparison Project – phase 6 (CMIP6, Eyring et al., 2016) focusing on the ice sheets. ISMIP6 is the first ice sheet model intercomparison that is fully integrated within CMIP. This is an improvement to earlier initiatives like ice2sea (Gillet-Chaulet et al., 2012; Shannon et al., 2013; Goelzer et al., 2013; Edwards et al., 2014a; b) and SeaRISE (Sea-level Response to Ice Sheet Evolution; Bindschadler et al., 2013; Nowicki et al., 2013a, b) that were lagging one iteration behind in terms of

applied climate forcing. More information on ISMIP6 can be found in the description paper (Nowicki et al., 2016) and on the Climate and Cryosphere (CliC) hosted webpage: (http://www.climate-cryosphere.org/activities/targeted/ismip6).

The initialisation of an ice sheet model forms the basis for any prognostic model simulation and therefore reveals most of the modelling decisions that distinguish different approaches (Goelzer et al., 2017). It consists of defining both the initial physical state of the ice sheet and model parameter values. In the context of initMIP-Greenland, we focus on initialisation to the present day as a starting point for centennial time-scale future sea-level change projections (Nowicki et al., 2016). The need for physical

ice flow models for such projections lies in the dynamic and highly nonlinear response of ice sheet flow to changes in climatic forcing at the atmospheric and oceanic boundaries. The surface mass balance (SMB) of the ice sheet is governed by the amount of precipitation falling on the surface and by meltwater runoff removing mass predominantly at the margins. Mass is also lost by melting at surfaces in contact with ocean water and by calving of icebergs from marine-terminating outlet glaciers. Changes

in ice sheet geometry generally cause changes in atmospheric conditions over the ice sheet, and hence changes in SMB. An important effect is the height-SMB feedback, which causes decreasing SMB with decreasing ice surface elevation and vice versa (e.g., Helsen et al., 2012; Franco et al., 2012; Edwards et al., 2014a; b). An important consequence of the relation between SMB and ice flow is that negative SMB removes ice before it can reach the marine margins and thereby reduces the calving flux (e.g.,

Gillet-Chaulet et al., 2012; Goelzer et al., 2013; Fürst et al., 2015). An estimate for the recent balance of processes indicates that ablation (i.e., negative SMB) is responsible for two-thirds of the increasing GrIS mass loss in the period 2009-2012, with ice discharge from marine-terminating outlet glaciers accounting for the remaining third (Enderlin et al., 2014). While the relative importance of outlet glacier dynamics for total GrIS mass loss has decreased since 2001 (Enderlin et al., 2014) and is expected to

decrease further in the future (e.g., Goelzer et al., 2013; Fürst et al., 2015), it remains an important aspect for projecting future sea-level contributions from the ice sheet on the centennial time scale.

Observations of ice sheet geometry and surface velocity, which ultimately form the target for any initialisation to the present-day state, exist only since the advent of the satellite era for the last ~25 years

(e.g., Mouginot et al., 2015). This is a short period compared to the longer response times of the ice sheet, that can be up to several thousand years (Drewry et al., 1992), and makes it impossible to understand ice sheet changes based on observations alone. While detailed observations mainly cover the ice surface properties, measurements for the ice interior and bed conditions are limited to a handful of deep ice core drilling sites. Radar layers dated at ice core sites can be used to extend the dating over

large parts of the Greenland ice sheet (MacGregor et al., 2015), but this information is not well explored by ice sheet modellers so far.

Projections of ice sheet response on decadal to centennial timescales are strongly influenced by the initial state of the ice sheet model (e.g., Arthern and Gudmundsson, 2010; Nowicki et al., 2013b;

Adalgeirsdottir et al., 2014; Saito et al., 2016). The prognostic variables and parameters that need to be defined for the initial state of an ice sheet model at the present day depend to some extent on the complexity of the modelling approach, but typically consist of ice temperature (due to its impact on

both ice rheology and basal slip), ice sheet geometry, and boundary conditions at the base of the ice sheet. For this timeframe, ice sheet modellers face an issue similar to the one encountered in the weather/climate community: whether to treat the problem as a "boundary value problem" (climate prediction) or as an "initial value problem" (weather forecasting).

Models developed for long-term and paleoclimate simulations typically use 'spin-up' procedures to determine the initial state, where the ice sheet model is run forward in time for tens to hundreds of thousands of years with (changing) reconstructed or modelled climatic boundary conditions (e.g., Huybrechts and de Wolde, 1999; Greve et al., 2011; Aschwanden et al., 2013). This implies that at any
time during the simulation (except at the beginning where arbitrary conditions are set), the model's state is defined as a consistent response to the forcing. Imperfections due to applied physical approximations, limited spatial resolution, and uncertainty in physical parameters and climatic boundary conditions can result in a considerable mismatch between the spun-up state and present-day observations.

The main alternative to the spin-up approach is to use data assimilation techniques, which leverage high-resolution observations of geometry and velocity to initialise ice sheet models to the present-day state (e.g., Gillet-Chaulet et al., 2012; Seroussi et al., 2013; Arthern et al., 2015). They typically infer poorly constrained basal conditions by a formal partial-differential-equations-constrained optimisation to match observed surface velocities for a given geometry (e.g., Morlighem et al., 2010). This implies
that the inferred basal parameters remain constant throughout the simulation, which is limited to the centennial time scale where this is approximately the case. Data assimilation techniques produce an initial state as consistent as possible with observational data, but are affected by inconsistences (e.g., ice temperature not in equilibrium with the stress regime) and by uncertainties in observations (e.g., inconsistencies between different observational datasets (Seroussi et al., 2011)). As data assimilations
are designed to best fit observations, errors arising from choices in ice parameters, physical processes, model resolution, observational data sets or from ignoring relevant state variables (e.g. ice rheology) are transferred to basal conditions or other parameters obtained by inversion. An intermediate approach is assimilation of the geometry of the ice sheet, by finding basal conditions that reduce the mismatch with the observed ice sheet surface (Pollard and DeConto, 2012b). This method is typically applied during
forward integration of the model and implies a model state in near balance with the forcing, though with a degree of compromise over matching observations. Note that the division of the different initialisation approaches presented here is somewhat arbitrary. Combinations between different approaches (e.g. relaxation after data assimilation) exist and need to be further explored to improve initialisation techniques in the future.

Given the wide diversity of ice sheet initialisation techniques, the goal of initMIP-Greenland is to document, compare and improve the techniques used by different groups to initialise their state-of-the-art whole-ice-sheet models to the present day as a starting point for centennial time-scale future sea-level change projections. A related goal is to highlight and understand how much of the spread in
simulated ice sheet evolution is related to the choices made in the initialisation. All three methods currently used for initialisation of ice sheet models (spin-up, assimilation of velocity, and assimilation of surface elevation) and variations thereof are represented in our ensemble. We first describe our

approach and experimental setup in Section 2, and present the participating models in Section 3. Section 4 concentrates on the results, with the ice sheet model initial state explored in Section 4.1, and the impact of initialization on ice sheet evolutions analysed in Section 4.2. Discussion and conclusions follow in Section 5.

## 2   Approach and Experimental setup

In initMIP-Greenland we focus on standalone ice sheet models, i.e., models not coupled to climate models. Although some participating models have the capability to produce their own SMB forcing, this is not a requirement in the present study. We have chosen to leave most of the modelling decisions to the discretion of the participants, which serves to document the current state of the initialisation techniques used in the ice sheet modelling community. Conversely, this implies a relatively heterogeneous ensemble with only incidental overlap of modelling choices between different submissions.

Experiments for the large-scale Greenland ice sheet have been designed to allow intercomparison of the modelled initial present-day states and of the model responses to a large perturbation in SMB forcing (Table 1). Modellers were asked to initialise their model to the present day with the method of their choice (*init*) and then run two forward experiments to evaluate the initialisation in terms of model drift: a control run without any change in the forcing (*ctrl*) and a perturbed run with a large prescribed surface mass balance anomaly (*asmb*). The prescribed SMB anomaly in experiment *asmb* (Appendix A) implies a strongly negative SMB forcing, in line with what may be expected from upper-end climate change scenarios. Nevertheless, the sea-level contribution from these experiments should not be interpreted as a projection, but rather as a diagnostic to evaluate model differences.

Note that the time of initialisation was not strictly defined (in the range 1950-2016), as modellers assign different dates to their initial state according to the data sets used. The participants were also largely free in other modelling decisions, with only the imposed constraint for the forward experiments that all boundary conditions and forcing remain constant in time. In particular, the SMB is not allowed to change (e.g., with surface elevation) other than by the prescribed SMB anomaly. All information and documentation concerning the ISMIP6 initMIP-Greenland experiments can be found on the ISMIP6 wiki (http://www.climate-cryosphere.org/wiki/index.php?title=InitMIP-Greenland).

While modellers were free to use a native model grid of their choice, model output was submitted on a common grid to support a consistent analysis (see Appendix C). This implies that results had to first be interpolated from the native model grid to the output grid, which for state variables has in most cases been done using conservative interpolation (Jones, 1999). In the following we present all results on the output grid with a horizontal resolution of 5x5 km. Furthermore, all ice sheet results have been masked to exclude ice on Ellesmere Island and Iceland.

**Table 1 Summary of the ISMIP6-initMIP-Greenland experiments**

| Experiment Title | Experiment label | CMIP6 Label (experiment_id) | Experiment Description | Duration of the simulation | Major Purposes |
|---|---|---|---|---|---|
| Initialisation | *init* | *ism-init-std* | initialisation to present day | n/a | Evaluation |
| Control | *ctrl* | *ism-ctrl-std* | unforced control experiment | 100 yr | Evaluation |
| SMB anomaly | *asmb* | *ism-asmb-std* | idealised change in SMB forcing | 100 yr | Evaluation |

## 3 Participating groups and models

Participants in initMIP-Greenland from 17 groups and collaborations (Table 2) have provided 35 model submissions. There is some overlap between the code bases used by different groups, with ultimately 11 individual ice flow models. However, the same model used by different groups (with varying datasets and initialisation procedures) may lead to rather different results. These submissions cover a wide spectrum of model resolutions, applied physical approximations, boundary conditions and initialisation techniques, which makes for a heterogeneous ensemble. In some cases, the same group has used two or more different model versions or different initialisation techniques, with several groups running their models at varying horizontal grid resolution. In the following we will refer to each separate submission as a 'model', identified by the model ID in the table of general model characteristics (Table 3). A detailed description of the individual models and initialisation techniques can be found in Appendix B.

**Table 2** Participants, ice sheet models and modelling groups in ISMIP6-initMIP-Greenland

| Contributors | Model | Group ID | Group |
|---|---|---|---|
| Victoria Lee, Stephen L. Cornford, Antony J. Payne, Daniel F. Martin | BISICLES | BGC | Centre for Polar Observation and Modelling, School of Geographical Sciences, University of Bristol, Bristol, UK / Department of Geography, College of Science, Swansea University, Swansea, UK / Computational Research Division, Lawrence Berkeley National Laboratory, Berkeley, California, USA |
| William H. Lipscomb, Joseph H. Kennedy | CISM | LANL | Los Alamos National Laboratory, Los Alamos, USA / National Center for Atmospheric Research, Boulder, USA / Climate Change Science Institute, Oak Ridge National Laboratory, Oak Ridge, USA / Computational Sciences and Engineering Division, Oak Ridge National Laboratory, Oak Ridge, USA |
| Fabien Gillet-Chaulet, Olivier Gagliardini | Elmer | IGE | Institut des Géosciences de L'Environnement, Univ. Grenoble Alpes, CNRS, IRD, Grenoble INP, IGE, F-38000 Grenoble, FR |
| Sainan Sun, Frank Pattyn | FETISH | ULB | Laboratoire de Glaciologie, Université Libre de Bruxelles, Brussels, BE |
| Philippe Huybrechts, Heiko Goelzer | GISM | VUB | Vrije Universiteit Brussel, Brussels, BE |
| Sébastien Le clec'h | GRISLI | LSCE | LSCE/IPSL, Laboratoire des Sciences du Climat et de l'Environnement, CEA-CNRS-UVSQ, Gif-sur-Yvette, FR |
| Fuyuki Saito, Ayako Abe-Ouchi | IcIES | MIROC | Japan Agency for Marine-Earth Science and Technology, JP / The University of Tokyo, Tokyo, JP |

| | | | |
|---|---|---|---|
| Heiko Goelzer, Roderik van de Wal | IMAUICE | IMAU | Utrecht University, Institute for Marine and Atmospheric Research (IMAU), Utrecht, NL |
| Helene Seroussi, Nicole Schlegel | ISSM | JPL | Caltech's Jet Propulsion Laboratory, Pasadena, USA |
| Helene Seroussi, Mathieu Morlighem | ISSM | UCI_JPL | Caltech's Jet Propulsion Laboratory, Pasadena, USA / University of California Irvine, USA |
| Martin Rückamp, Angelika Humbert | ISSM | AWI | Alfred Wegener Institute for Polar and Marine Research, DE / University of Bremen, DE |
| Andy Aschwanden | PISM | UAF | Geophysical Institute, University of Alaska Fairbanks, USA |
| Nicholas R. Golledge | PISM | ARC | Antarctic Research Centre, Victoria University of Wellington, NZ |
| Christian Rodehacke | PISM | DMI | Danish Meteorological Institute, DK / Alfred Wegener Institute for Polar and Marine Research, DE |
| Florian A. Ziemen | PISM | MPIM | Max Planck Institute for Meteorology, DE |
| Ralf Greve | SICOPOLIS | ILTS | Institute of Low Temperature Science, Hokkaido University, Sapporo, JP |
| Ralf Greve, Reinhard Calov | SICOPOLIS | ILTS_PIK | Institute of Low Temperature Science, Hokkaido University, Sapporo, JP / Potsdam Institute for Climate Impact Research, Potsdam, DE |

Despite the diversity in modelling approaches (Table 3) and the overlap between different methods, it is useful to distinguish the three main classes of initialisation techniques described before: first, those using a form of data assimilation (DA) to match observed velocities (DAv); second, those that rely

solely on model spin-up (SP), and third, the intermediate case of transient assimilation to match surface elevation (DAs). However, even DAv is typically preceded by some form of spin-up (with the same model or a different one) to produce the internal temperature of the ice sheet, and may also be followed by a relaxation run to make the velocities and geometry more consistent. The represented cases of DA infer a spatially varying basal drag coefficient to minimise the mismatch with observations of velocity

or geometry. Models using SP use physical parameters and processes to define the basal conditions.

Modelling choices also differ based on model purpose and typical application. Many of the SP models have been built and used for paleo applications for time periods when possible DA targets are very limited and SMB boundary conditions differ from the present. This makes it necessary in those models

to parameterise SMB, e.g., by using positive-degree-day (PDD) models (e.g., Huybrechts et al., 1991). SP approaches are also generally favoured when including ice sheets in coupled climate models. In two groups (DMI, MPIM), the ice sheet models and SMB forcing are set up in a similar way as they would be for coupled ice sheet-climate simulations. In contrast, the DAv models are built specifically for centennial time-scale future projections, while DAs again represents an intermediate case of models

typically used for long-term simulations, but specifically initialised for the present day. These fundamental differences in modelling approaches have to be kept in mind when comparing the models. The SMB is in many cases taken from regional climate model (RCM) simulations, but arises in some cases from parameterisations based on the modelled ice sheet geometry applying traditional PDD methods.

**Table 3** Model characteristics

Numerical method: FD= Finite difference, FE= Finite element, FV= Finite Volume with adaptive mesh refinement

Ice flow: SIA= Shallow ice approximation, SSA= Shallow shelf approximation, HO= Higher order, HYB= SIA and SSA combined
Initialisation method: DAv= Data Assimilation of velocity, DAs= Data Assimilation of surface elevation, SP= Spin up
Initial SMB: RA1= RACMO2.1, RA3= RACMO2.3, HIR= HIRHAM5, MAR= MAR, BOX= BOX reconstruction (synthesis of simulation and data), PDD= Positive Degree Day Model, EBM= Energy Balance Model (EBM)
Velocity: RM= Rignot and Mouginot, J= Joughin et al.
Bed and surface: M= Morlighem et al., B= Bamber et al., H= Herzfeld et al.
Geothermal Heat Flux (GHF): SR= Shapiro and Ritzwoller, G= Greve, P= Purucker, FM= Fox Maule et al., CST= Constant
Model resolution (Res) in km. In case of heterogeneous grid resolution, the minimum and maximum resolution are given.

| Model ID | Numerics | Ice flow | Initialisation | Initial year(s) | Initial SMB | Velocity | Bed | Surface | GHF | Res min | Res max |
|---|---|---|---|---|---|---|---|---|---|---|---|
| ARC-PISM | FD | HYB | SP | 2000 | RA1 | | B | | SR | 5 | 5 |
| AWI-ISSM1[*] | FE | HO | DAv | 2000 | RA3 | RM | M | | SR | 2.5 | 35 |
| AWI-ISSM2[*] | FE | HO | DAv | 2000 | RA3 | RM | M | | SR | 2.5 | 35 |
| BGC-BISICLES1 | FV | SSA | DAv | 1997 - 2006 | HIR | RM | M | | | 1.2 | 4.8 |
| BGC-BISICLES2 | FV | SSA | DAv | 1997 - 2006 | HIR | RM | M | | | 2.4 | 4.8 |
| BGC-BISICLES3 | FV | SSA | DAv | 1997 - 2006 | HIR | RM | M | | | 4.8 | 4.8 |
| DMI-PISM1[+] | FD | HYB | SP | 2000 | PDD | | B | | SR | 5 | 5 |
| DMI-PISM2[+] | FD | HYB | SP | 2000 | PDD | | B | | SR | 5 | 5 |
| DMI-PISM3[+] | FD | HYB | SP | 2000 | PDD | | B | | SR | 5 | 5 |
| DMI-PISM4[+] | FD | HYB | SP | 2000 | PDD | | B | | SR | 5 | 5 |
| DMI-PISM5[+] | FD | HYB | SP | 2000 | PDD | | B | | SR | 5 | 5 |
| IGE-ELMER1 | FE | SSA | DAv | 2000 - 2010 | MAR | J | M | | | 1.5 | 45 |
| IGE-ELMER2 | FE | SSA | DAv | 2000 - 2010 | MAR | J | M | | | 1 | 5 |
| ILTS-SICOPOLIS | FD | SIA | SP | 1990 | PDD | | B | | P | 5 | 5 |
| ILTSPIK-SICOPOLIS | FD | SIA | SP | 1990 | PDD | | H | | G | 5 | 5 |
| IMAU-IMAUICE1 | FD | SIA | SP | 1990 | RA3 | | B | | SR | 5 | 5 |
| IMAU-IMAUICE2 | FD | SIA | SP | 1990 | RA3 | | B | | SR | 10 | 10 |
| IMAU-IMAUICE3 | FD | SIA | SP | 1990 | RA3 | | B | | SR | 20 | 20 |
| JPL-ISSM | FE | SSA | DAv | 2012 | BOX | RM | M | | SR | 1 | 15 |
| LANL-CISM | FE | HO | SP | 1961 - 1990 | RA1 | | M | | CST | 4 | 4 |
| LSCE-GRISLI | FD | HYB | DAv | 2000 | MAR | J | B | | FM | 5 | 5 |
| MIROC-ICIES1 | FD | SIA | DAs | 2004 | RA1 | | B | B | SR | 10 | 10 |
| MIROC-ICIES2 | FD | SIA | SP | 2004 | PDD | | B | | SR | 10 | 10 |
| MPIM-PISM | FD | HYB | SP | 2006 | EBM | | B | | SR | 5 | 5 |
| UAF-PISM1[#] | FD | HYB | SP | 2007 | RA1 | | M | | SR | 1.5 | 1.5 |
| UAF-PISM2[#] | FD | HYB | SP | 2007 | RA1 | | M | | SR | 3 | 3 |
| UAF-PISM3[#] | FD | HYB | SP | 2007 | RA1 | | M | | SR | 4.5 | 4.5 |
| UAF-PISM4[#] | FD | HYB | SP | 2007 | RA1 | | M | | SR | 1.5 | 1.5 |
| UAF-PISM5[#] | FD | HYB | SP | 2007 | RA1 | | M | | SR | 3 | 3 |
| UAF-PISM6[#] | FD | HYB | SP | 2007 | RA1 | | M | | SR | 4.5 | 4.5 |
| UCIJPL-ISSM | FE | HO | DAv | 2007 | RA1 | RM | M | | SR | 0.5 | 30 |
| ULB-FETISH1 | FD | SIA | DAs | 1979 - 2006 | MAR | | B | B | FM | 10 | 10 |
| ULB-FETISH2 | FD | HYB | DAs | 1979 - 2006 | MAR | | B | B | FM | 10 | 10 |
| VUB-GISM1 | FD | HO | SP | 2005 | PDD | | B | | SR | 5 | 5 |
| VUB-GISM2 | FD | SIA | SP | 2005 | PDD | | B | | SR | 5 | 5 |

*AWI-ISSM2 differs from AWI-ISSM1 in the climatic forcing used during temperature spin-up. [+]DMI-PISM1-5 differ in the melt parameters of the PDD model. [#]UAF-PISM4-6 differ from UAF-PISM1-3 in the initial geometry.

## 4    Results

In this section, we first present results of the *init* experiment, designed to compare the present-day initial state between participating models and against observations. These or similar initial model states would serve as a starting point for physically-based projections of the Greenland ice sheet contribution to

future sea-level changes (Nowicki et al., 2016). We then present results for the two forward experiments that serve to further evaluate the response of these initial states to idealised forcing (*ctrl*, *asmb*).

## 4.1    Evaluation of the initial state

Because initialisation techniques generally differ in the observational data used as model input,
boundary conditions and assimilation targets, we did not prescribe the year(s) of initialisation. The initialisation times in the ensemble (Table 3) therefore represent the time frame(s) of the observations that are used for data assimilation (in case DA) and the simulated SMB used as boundary condition for the individual models. For the comparative analysis, we did not attempt to correct the differences arising from different initialisation times. Compared to the range of modelling uncertainties, this
assumption probably holds for the geometry of the ice sheet, but is more questionable for velocity. However, the sparseness and limited temporal resolution of available observations excludes analysing models with respect to their individual reference time frame. Where available, we have used a range of observational data sets to compare against.

The modelled present-day ice extent (Figure 1) exhibits a large spread among models and ranges from the extent of the observed ice sheet proper (excluding connected glaciers and ice caps) to nearly filling the entire land above sea level (see also supplementary Figure S2 for individual model results). This diversity in the ensemble is representative of the large range of modelling choices and initialisation techniques. For example, the assumption of what should be modelled (only the ice sheet, or including
outlying glaciers and ice caps) differs from group to group. Also, models may simulate ice in places where no ice is observed. While some models prescribe a fixed (observed) ice mask and prevent any ice growing outside, most models simulate ice margins that are free to evolve according to the balance of ice flow and SMB. In some cases, modellers have controlled the extent where ice sheets are allowed to grow, e.g., by prescribing a negative SMB over observed ice-free regions.

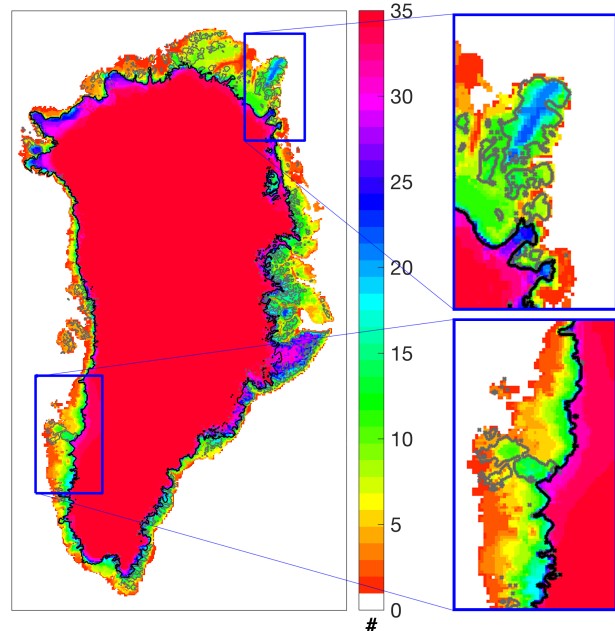

**Figure 1 Common ice mask of the ensemble of models in the intercomparison. The colour code indicates the number of models (out of 35 in total) that simulate ice at a given location. Outlines of the observed ice sheet proper (Rastner et al., 2012) and all ice-covered regions (i.e. main ice sheet plus small ice caps and glaciers; Morlighem et al. 2014) are given as black and grey contour lines, respectively. A complete set of figures displaying individual model results is given in the supplementary material.**

The diversity of modelling choices equally leads to a large spread in the simulated total ice area and volume at the present day (Figure 2, see supplementary Table S1 for the numbers). In comparison with observations (Morlighem et al., 2014), the initial ice sheet area (horizontal axis in Figure 2) of many models is 'bracketed' by the observed extent (cf. Figure 1) of the ice sheet proper (black diamond) and the extent of the entire ice-covered areas (grey diamond). Differences in observed volume (vertical axis in Figure 2) between these two defined areas are small compared to the ensemble spread, i.e. the proportional change from including ice caps and glaciers surrounding the Greenland ice sheet to volume (0.3 %) is much smaller than to area (8.2 %). An alternative data set (Bamber et al., 2013) provides similar numbers for observed volume and area (not shown). Overestimation of modelled ice sheet area (by up to 15%) is common, and overestimation of volume (up to 15%) is more prevalent than underestimation.

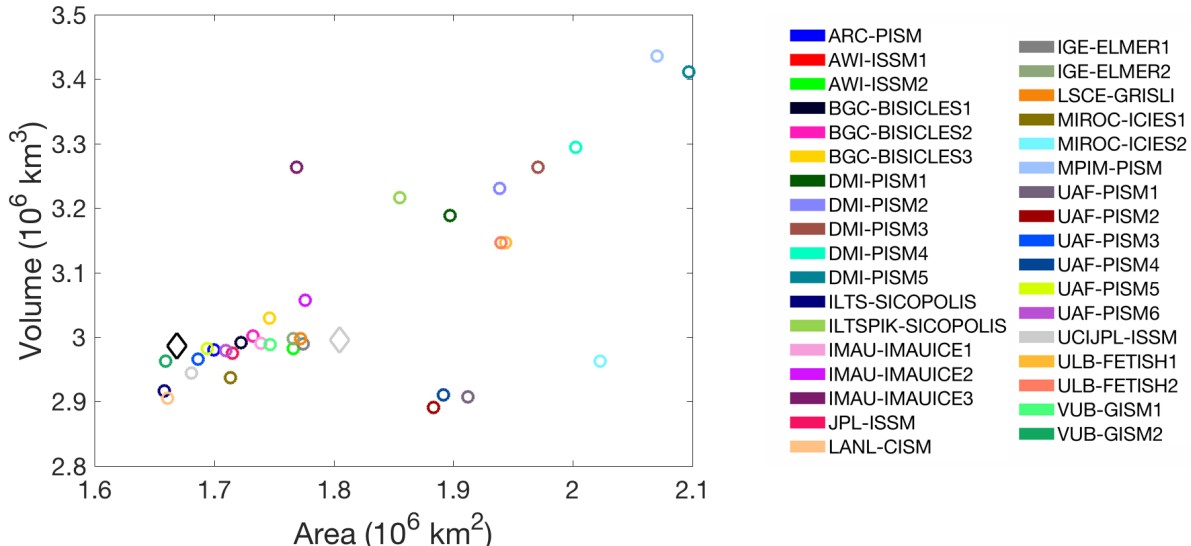

**Figure 2 Grounded ice area and grounded volume for all models (circles). Observed values (Morlighem et al. 2014) are given for the entire ice covered region (grey diamond) and for the region of the ice sheet proper (black diamond) according to the mask of Rastner et al. (2012) shown in Figure 1.**

As an important input for the ice sheet simulations, we evaluate the implemented SMB of the different models. Figure 3 shows the typical present-day SMB applied for three of the models spanning the distribution of ice thickness error (see below), while an overview of all models is given in supplementary Figure S2. In the three cases shown, one model applied SMB from a RCM with no

10 modification (AWI-ISSM2), another (MIROC-ICIES) used a PDD method and the last (MPIM-PISM) obtains the SMB from an energy balance model (EBM) designed for coupling of the ice sheet model to a climate model.

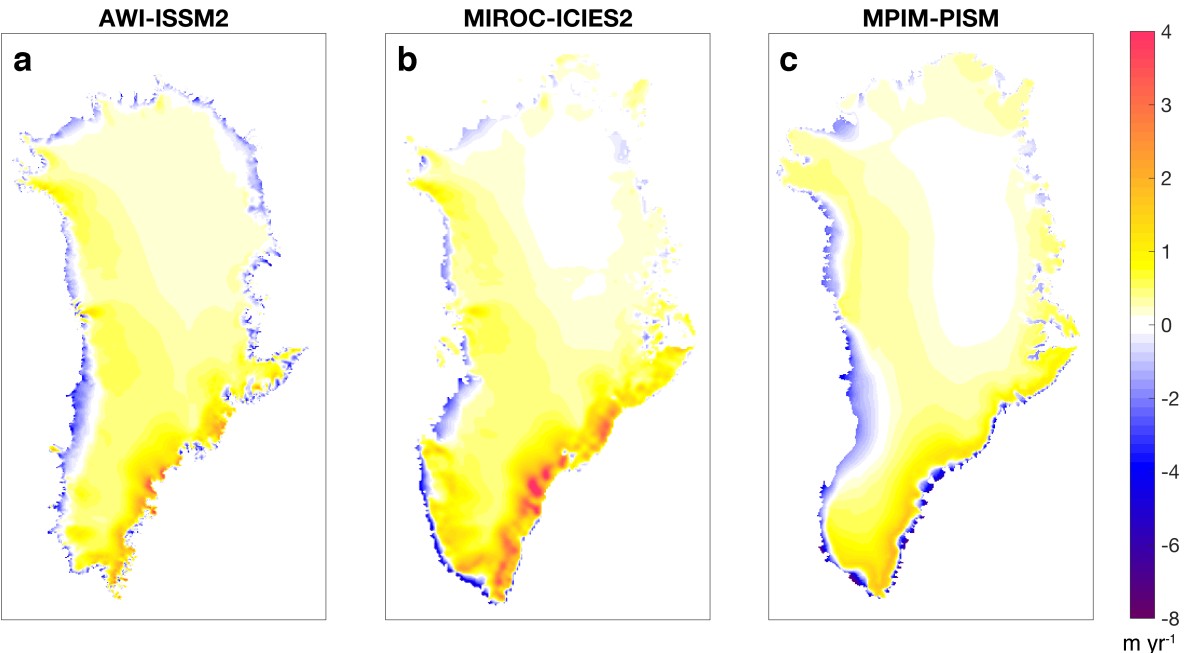

**Figure 3 Typical surface mass balance for the initial state for three different models. Note the unequal scaling for positive and negative values.**

Because we generally cannot distinguish individual accumulation and ablation processes for the SMB prescribed during initialisation, we separate the assumed SMB into negative and positive regions (i.e., ablation and accumulation zones) for further analysis. Partitioning of mass change processes between SMB and dynamic changes (e.g., van den Broeke et al., 2009; Enderlin et al., 2014) would also be an important diagnostic to analyse. However, the participating models have not incorporated the required

mechanisms, and we also lack the necessary forcing, to generate fast dynamical response due to outlet glacier changes in the current experiments. Displaying the SMB magnitude for accumulation and ablation regimes allows us to identify some important outliers (Figure 4a) and frame the model input compared to estimates of total SMB from a range of RCMs (Figure 4b). Apparent outliers are models with small ablation zones and large positive SMB (far right in Figure 4a) and those with a large ablation

area (top in Figure 4a). Several of the remaining models cluster around RCM estimates (van Angelen et al., 2014; Fettweis et al., 2017; Noël et al., 2016) for the SMB partitioning, again considering either all ice-covered regions or only the ice sheet proper. This is mostly the case because the models use these or similar products. However, an additional condition required for close agreement with RCM estimates is that the modelled ice sheet is close to the observed extent. Models that lie further from RCM estimates

(in Figure 4b) typically have larger ablation zones and consequently larger negative SMB.

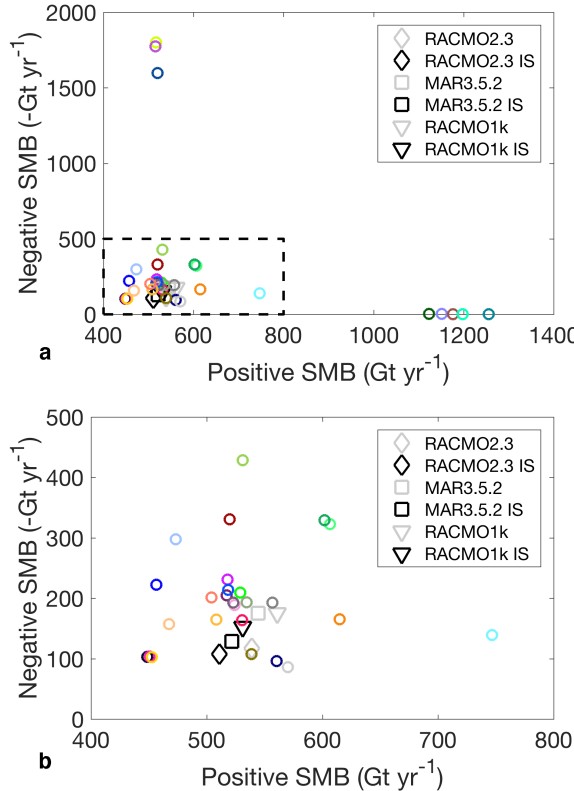

**Figure 4 Negative and positive SMB of all models (a) and for the marked inset excluding outliers (b). Diamonds, squares and triangles in (b) give partitioning from average 1979-2000 regional climate model simulations (van Angelen et al., 2014; Fettweis et al., 2017, Noël et al., 2016) with masking to the ice-covered region (grey) and to the ice sheet proper (IS, black) according to the mask of Rastner et al. (2012). Compare symbol colour to identify individual models with Figures 3-5. Data are available in supplementary Table S1.**

We further evaluate the prescribed SMB in comparison to point observations (Figure 5). Available SMB observations (Machguth et al., 2015; Bales et al., 2009) are sparse, especially in the centre of the ice sheet, and have heterogeneous temporal coverage. However, comparison against those observations allows for a first-order evaluation of the SMB inputs chosen or produced by the modellers. Overall, positive SMB is better represented in the chosen SMB datasets than negative SMB. The order-of-magnitude difference in root mean square error (RMSE) between the two measures is partly explained by the relatively low accumulation over a large area in the centre of the ice sheet, compared to relatively high ablation over a narrow marginal zone, which is easily misrepresented in models with too large an ice extent. While the best match with observations in both regions is produced by models using SMB derived from RCMs, good agreement with the observed SMB can also be found for some models using PDD. Again, a good match with the observed ice extent is more important than the SMB model itself to reduce the mismatch with measured SMB.

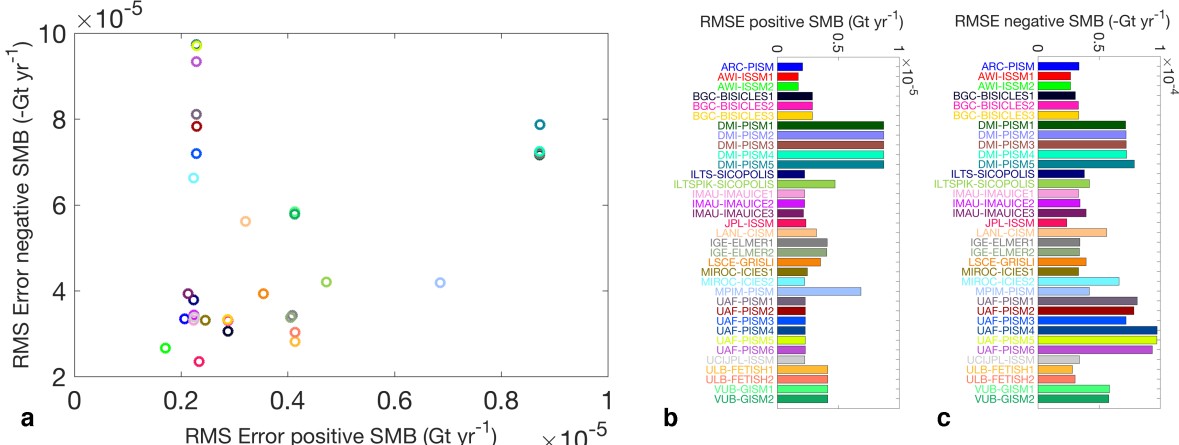

**Figure 5 Root mean square error (RMSE) of initial modelled surface mass balance compared to observations for the accumulation zone (b and abscissa in a) and the ablation zone (c and ordinate in a). The observational data sets are from Bales et al. (2009) for the accumulation zone and Machguth et al. (2015) for the ablation zone.**

The match of the initial ice sheets with observed geometry (Morlighem et al., 2014) is evaluated as the RMSE between modelled and observed ice thickness (Figure 6). Interpretation of the diagnostics requires distinction between the different initialisation techniques, because the geometry is a prognostic variable in some cases of SP, but a given constraint during initialisation for DA. In some cases of SP, the ice sheet area is effectively confined to the observed, which represents an intermediate case. For models covering a range of horizontal resolutions, accuracy generally decreases with coarser horizontal grid resolution (BISICLES, IMAUICE, ELMER), except for UAF_PISM, where the trend is not clear. Using a different observational data set (Bamber et al., 2013) to calculate the diagnostic gives overall similar results (not shown). However, it is noticeable that DA models that have been initialised with one data set show lower errors when comparing with that specific set of observations. This point requires attention, should this diagnostic be used to formally evaluate and score the models at some stage.

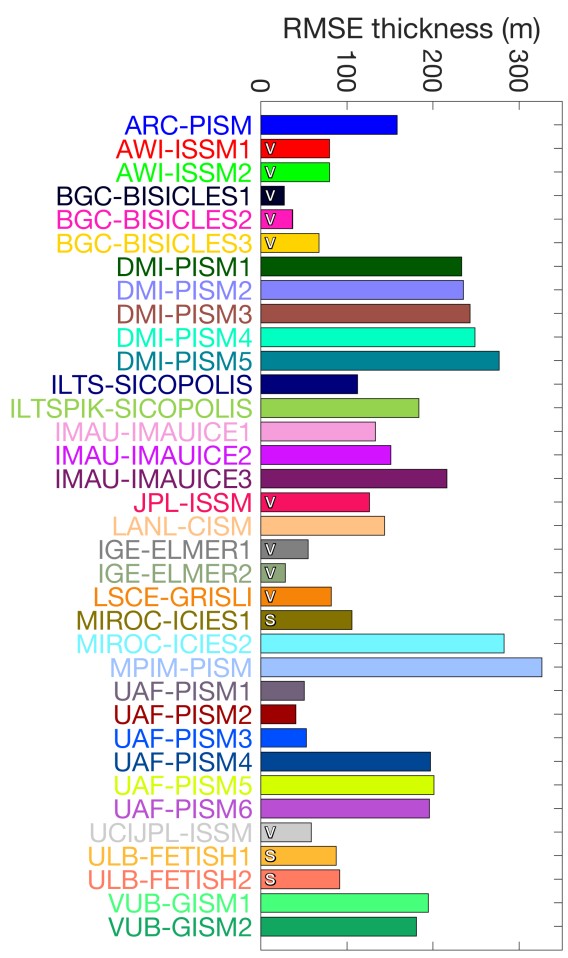

**Figure 6 Root mean square error (RMSE) of initial modelled ice thickness compared to observations (Morlighem et al. 2014). The diagnostic has been calculated for subsampled data to reduce spatial correlation, and we show median values for different offsets. Letters in the bars denote assimilation targets for methods DAv and DAs and are left empty for SP.**

To evaluate the match of the models with observed surface ice velocities, we have computed the RMSE between the modelled and observed (Joughin et al., 2016) velocity magnitude (Figure 7a). Calculating the RMSE based on the log of the velocities instead (Figure 7b) results in a slightly different picture, because errors in high velocities typically occurring at the margins over a relatively small fraction of the
10 ice sheet area are weighted less. We note that an alternative choice of metric would be one that accounts for spatial variation in observational uncertainty, such as standardised Euclidean distance. Distinction between models using DAv and the rest is again useful, since velocity is not an independent variable in cases where it enters the inversion calculations. Models using observed velocities in the DAv procedure

could in principle be compared with each other to evaluate the success of the inversion technique. However, the comparison would have to take into account that some groups use relaxation after the DAv step to get a better consistency between the ice geometry and velocity. This modifies the results depending on the relaxation time. Better consistency for a model can be achieved with longer relaxation
time, at the expense of a larger discrepancy with the observed geometry. In any case, not every group uses the same velocity dataset (e.g., Rignot and Mouginot, 2012; Joughin et al., 2016).

It is interesting to note that DAs techniques using only surface elevation as inversion target can have quite low errors in simulated velocities, which implies an overall consistency between geometry and
velocity structure of the modelled ice sheets. Although this consistency is expected based on mass conservation, the results confirm that the basic assumptions (e.g., approximation to the force balance and rheology structure) are generally close enough to reality. This is particularly important considering that DAv techniques can match observed velocities well for almost any given rheology, as all the uncertainty (including unknown rheology) is compounded in the basal sliding relation.

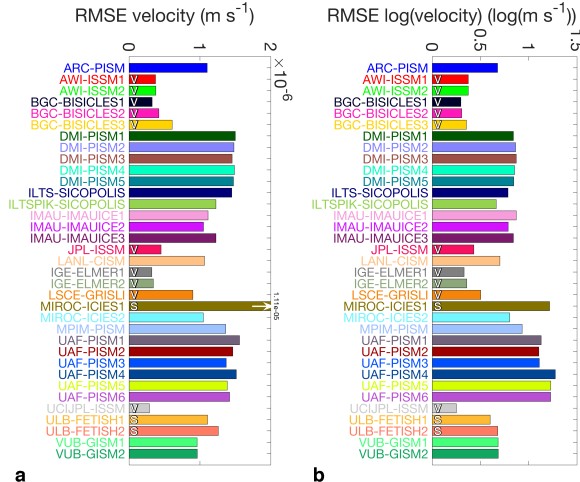

**Figure 7 Root mean square error (RMSE) of the horizontal velocity magnitude (a) and the log of the horizontal velocity magnitude (b) compared to observations (Joughin et al., 2016). The diagnostics have been calculated for grid cells subsampled regularly in space to reduce spatial correlation; we show median values for different possible offsets of this sampling.**

**4.2      Results of the forward experiments**

The two experiments *ctrl* and *asmb* have been performed to further test the modelled initial states in terms of their behaviour in typical forward simulations. This is needed to expose model response to changing constraints that were present during initialisation. Furthermore, we evaluate the influence of the initial state and of modelling decisions pertaining to the initialisation on the results of the forward
experiments, i.e., the projected ice thickness change and sea-level contribution.

The experiment *ctrl* serves to evaluate the model response in the absence of additional forcing and is an important step to understand the consequence of modelling choices for forward experiments. Since we have not specified any assumption on the imbalance between SMB and ice flow at the initial state, the ice sheet would ideally exhibit an imbalance that matches observations for a given time interval. Recent

modelled mass trends or thickness changes could then in principle be evaluated with existing observational data sets of limited time coverage (e.g., Velicogna et al., 2014). Reproducing recent changes seen by GRACE (mass change) and altimetry (thickness change), however, is hampered by not knowing the ice sheet bedrock and surface elevation well at the time that the satellite started to observe, and would also assess the accuracy of the SMB input (i.e., for many models, a separate RCM).

Furthermore, this would require that the experiments aimed for realistic outlet glacier dynamics and ocean forcing (e.g., Nick et al., 2013), which is currently not available (Alexander et al., 2016; Schlegel et al., 2016) and has deliberately not been included in the present experiments. Approaches to validate models using hindcasting techniques (Aschwanden et al., 2013; Larour et al., 2014; 2016; Price et al., 2017) currently suffer the same limitations of observational data sets with short time coverage,

uncertainty in external forcing, limited knowledge of processes responsible for dynamic outlet glacier response and the initialisation problems discussed above.

The simulated ice mass evolution in *ctrl* (Figure 8a) reflects the wide spread of initial ice mass among the models and a relatively small mass change for most of them over the course of the 100-year

experiment. This is because a common approach is to attempt initialisation to a steady state with a given SMB forcing, possibly followed by relaxation or by a run with recent SMB forcing. Total mass changes in experiment *ctrl* (Figure 8b) range from ~-20 to +25 mm sea-level equivalent (SLE) when nine obvious outliers (discussed in context of Figure 4) are ignored. Note that the total mass change is not a complete measure of the model drift, since positive and negative trends at different places can

compensate. To calculate the SLE contribution for all models consistently, we have masked out ice outside of Greenland (Ellesmere Island and Iceland), considered only the ice mass above floatation, applied a correction to compensate for the map projection error (Snyder 1987) and converted volume to mass using the specific ice densities from each model. In some cases of the ensemble (typically for the SP models), the modelled background trend arises from transient forcing of SMB and temperature over

the past, but more often it is due to inconsistencies introduced during initialisation (i.e., the trends are dominated by the model's response to the initialisation, not to the forcing).

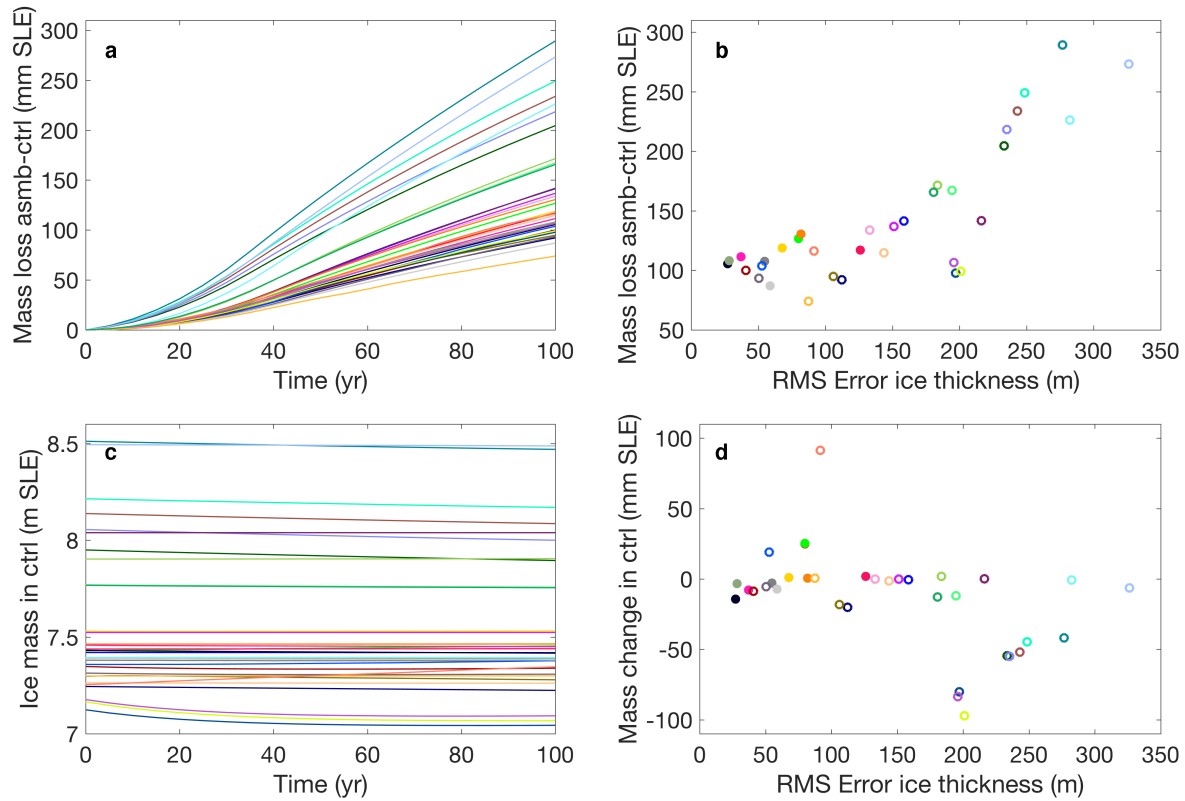

**Figure 8 Ice mass evolution in *ctrl* (a) and ice mass loss from *asmb-ctrl* (c). Mass change after 100 years in experiment *ctrl* (b) and from *asmb-ctrl* (d) related to error in initial ice thickness. Ice volume changes have been converted to sea-level equivalent (SLE) assuming an ocean area of 361.8 *10^6 km^2 and the specific ice density from the individual ice sheet models. Filled symbols in (b) and (d) represent DAv models. Data in (b) and (d) is available in supplementary Table S1.**

For DAv models (filled symbols in Figure 8b), the mass trend in experiment *ctrl* represents an important diagnostic to complement the measured accuracy of matching the observed geometry, because it will also reflect any inconsistencies between ice velocity and geometry datasets (Seroussi et al., 2011). This can be illustrated by considering a forward model run that is started from an exactly specified geometry as initial state. Optimized model velocities combined with imperfect ice thickness reconstructions may result in a flux divergence that is unbalanced by the local SMB, which leads to large model drift. Conversely, ice sheet models can be relaxed to a steady state when constraints on the target geometry are loosened completely. Match with the observed geometry in the initial state and model drift in the forward experiment are therefore complementary measures that should be considered together. While this is evident for any single model, we only find tentative confirmation amongst the DAv models in our ensemble (filled symbols in Figure 8b), with increasing mass trend for decreasing ice thickness error.

The simulated sea-level contribution of the models, calculated from the difference in mass change between *asmb* and *ctrl,* shows a large spread of 75 to 290 mm SLE (Figure 8c, d), indicative of the wide range of modelled ice sheet extent (and therefore ice thickness error). This relation arises because the

prescribed SMB anomaly has been optimized for the observed geometry, but has not been limited to the observed ice sheet extent. The typical SMB field at the end of experiment *asmb* (illustrated for three different models in Figure 9) is strongly negative along the ice sheet margin, with an ablation zone that covers all of the ice sheet margin and extends several hundred kilometres inland in the southwest and
northeast of Greenland. For models with (unrealistically) large initial (present-day) surface areas, the ablation zones are considerably larger (Figure 9b, c), which implies dramatic mass loss. The too large ice sheet area is related to the definition of the ice sheet with respect to outlying glaciers and, more importantly, due to modelled initial conditions further from the present-day. Simply put, by design a larger ice sheet will be subject to larger rates of mass loss.

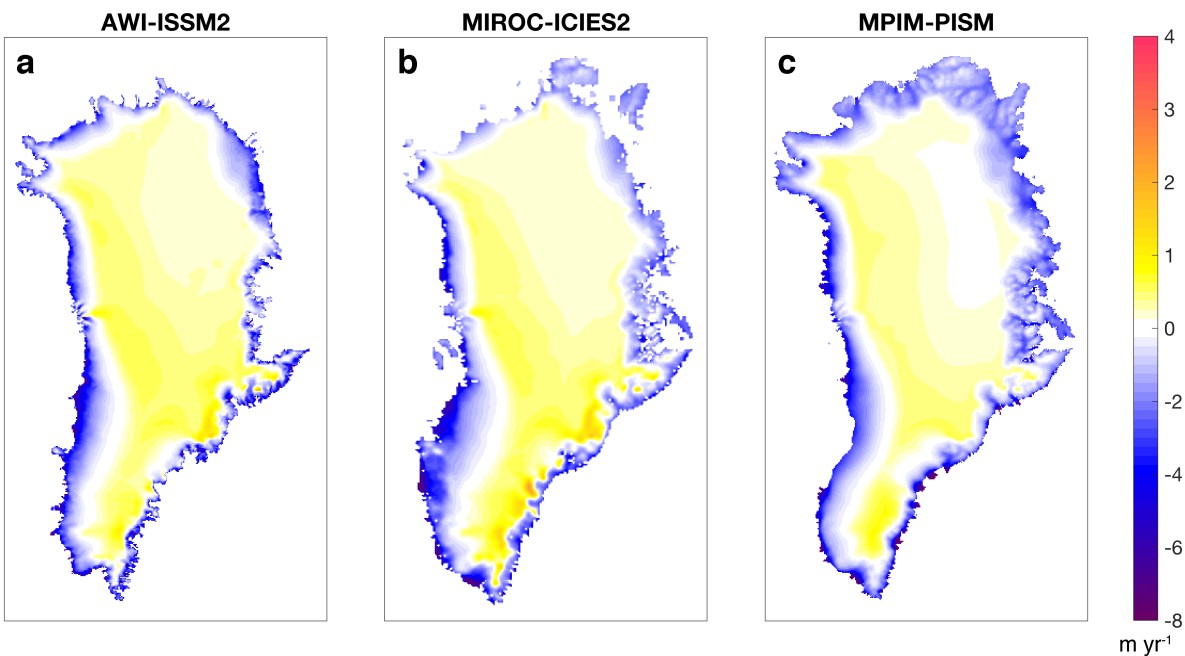

**Figure 9 Typical surface mass balance after 100 years in experiment *asmb* for the three models in Figure 3. Note the unequal scaling for positive and negative values.**

The spatial patterns of thickness changes in experiment *ctrl* (Figure 10a, b, c and Figure S5) clearly reflect some important differences and similarities between the models and used initialisation techniques. DA models typically exhibit more noise (e.g. Figure 10a) compared to SP models (e.g. Figure 10b, c), which arises from inconsistencies between geometry and velocity for the former as discussed above. Models with identical model setup, but at different horizontal resolution show similar
patterns, and the same applies for different versions of one model (DMI-PISM), which differ only by the PDD parameters (Figure S5). In all cases, thickness changes are the largest close to the margin and less pronounced in the interior of the ice sheet, a difference that becomes clearer with longer relaxation time. The patterns also confirm that positive and negative thickness changes at different locations often compensate each other so that the total mass change in experiment *ctrl* (Figure 8c and Table S1) is not a
complete measure of the model drift. Because the thickness change in experiment *ctrl* is mostly due to

unwanted model drift, we have calculated the mass evolution (Figure 8c) and sea-level contribution (Figure 8d) from ice thickness change differences between *asmb* and *ctrl* (Figure 10d, e, f and Figure S6). This is a common workaround to remove model drift and facilitate model comparison, but it also neglects the contribution of any prognostic imbalance and present-day ice sheet evolution in the

5 resulting figures. In the centre of the ice sheet, the modelled thickness change (Figure 10d, e, f) is dominated by the prescribed SMB anomaly and therefore similar between all models (Figure S6), while marginal changes show again much larger differences.

In contrast to the large differences in modelled ice volume changes, which may largely be explained by

10 differences in initial ice sheet extent, we find that models are similar in the dynamic response within the region of overlap, i.e., within most of the observed ice mask. For this analysis, we have calculated the difference between modelled ice thickness changes (*asmb-ctrl*) and the time-integrated SMB anomaly for each individual model (see Figure 10g, h and i for three examples and Figure S7 for all models). This diagnostic, first shown and discussed by Huybrechts et al., (2002), represents ice thickness changes

15 due to the flow of the ice in response to changes in SMB: in other words, the extra information gained by using ice dynamic models over projections of SMB changes alone.

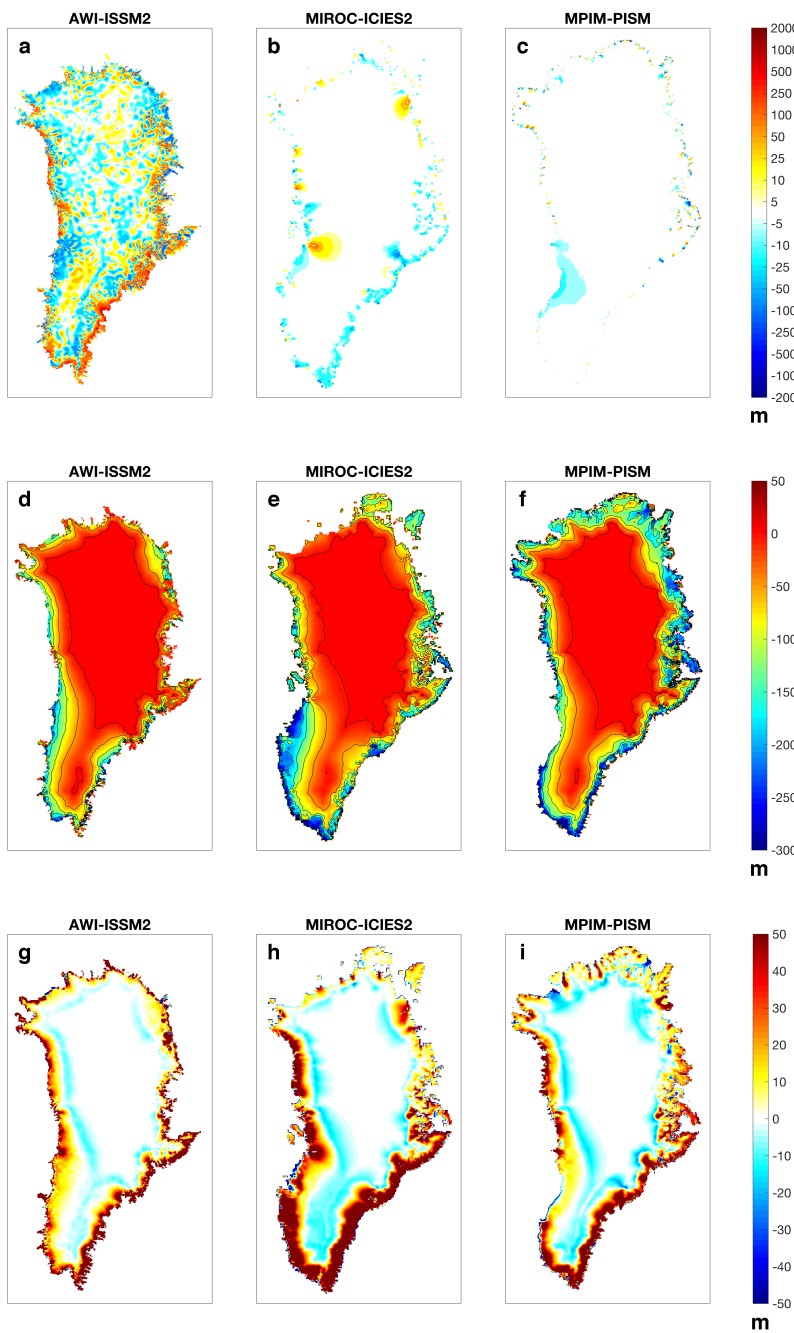

**Figure 10 Ice thickness change in *ctrl* after 100 years (top), difference of ice thickness change (*asmb-ctrl*) after 100 years (middle) and dynamic contribution (bottom) for the three models in Figure 3 and Figure 9. Note the nonlinear contour intervals in the top row.**

Dynamic thickening happens in regions of steep gradients in negative SMB anomalies around the margins of the ice sheet. Dynamic thinning occurs across the line separating positive and negative SMB anomalies, close to the equilibrium line. This pattern of dynamic response is reproduced by all models (see Figure S7), and shows strong similarities for the region of overlap across the entire ensemble. In other words, the models largely agree in their representation of the ice dynamical response to the applied SMB-anomaly forcing.

## 5    Discussion and conclusion

We have compared different initialisation techniques used in the ice sheet modelling community across a representative spectrum of approaches. While long-term processes and adjustment of internal variables (e.g., ice temperature and rheology) can be incorporated with SP methods, this occurs at the expense of a better match with observations of present-day ice sheet geometry and velocity and hence, the initial dynamic state of the ice sheet. Conversely, the initial states produced by the DAv approach exhibit a much better match with observations, at the expense of including long-term processes. The DAs method and other approaches to incorporate DA elements in SP models and vice versa form an intermediate group. At present, none of the methods is capable of combining both aspects (good match with observations and long-term continuity) sufficiently well that it would render other methods obsolete for all applications.

DAv is the method of choice for short-term projections with anomaly forcing, and as far as long-term dynamical interactions (e.g., arising from interaction with the basal conditions, the bedrock or from thermo-mechanical coupling) can be neglected. For long-term projections of ice sheet behaviour, where these interactions become important, SP and DAs methods are needed. The range of time-scales where this is the case is not well-defined and may lie anywhere between several decades and several centuries. For the standalone ice sheet projections planned for CMIP6 within ISMIP6 (100 to 200-year time scale), a combination of SP and DA methods may be required to simulate the response of the Greenland ice sheet to future climate change. The challenge remains how to initialise models to closely match the observed dynamical state and at the same time minimise unrealistic transients and incorporate the long-term evolution of thermodynamics and bedrock changes. A promising approach is additionally optimising the basal topography within observational errors as part of the data assimilation procedure (Perego et al., 2014; Mosbeux et al., 2016). Other approaches, based on assimilation of time-series of observed surface velocity (Goldberg et al., 2013) or surface elevations (Larour et al., 2014) exist for smaller scales, but should be further explored to eventually be applied over the entire ice sheet. A so far underexplored possibility is to use existing information from radar layers (MacGregor et al., 2015) as additional constraints in initialisation methods.

The present 'come-as-you-are' approach is well suited to produce an overview of initialisation techniques in the community and to compare individual models against observations. However, we have encountered difficulties in comparing models because of the wide variety of approaches. Differences in the initial ice sheet extent have rendered the locally identical SMB anomaly forcing to be different on

the global scale. We have found that estimating mass changes consistently across all models becomes a non-trivial undertaking, considering differences in ice sheet masks, projection errors and differences in model ice density. Additional problems arise from the use of different native grids (unstructured and structured) with potential artefacts introduced by interpolation that we have not been able to quantify.

The mismatch between observed and modelled ice sheet extent also needs an urgent solution in view of constructing an ensemble of sea-level change projections based on CMIP6 climate model data. The large ensemble spread in sea-level contribution in the *asmb* experiment is mostly due to the "extra" ice in the initial ice sheet geometry. At this stage, it is not clear how to minimize the contribution to sea level change due to this bias introduced solely by the experimental setup. Letting each model estimate its own SMB anomaly relative to the individual ice sheet geometry would likely reduce this problem, but it would complicate any further comparison by removing the constraint of locally identical SMB for all models.

Compared with earlier ice sheet model intercomparison exercises that have initialised ice sheet models for future projections (Bindschadler et al. 2013; Nowicki et al. 2013b), we find considerably less drift in the control experiments (for models that also participated in previous intercomparisons). We attribute this improvement to more attention of modellers on ice sheet initialisation and to an improved understanding of what is needed to achieve that goal, including the development of improved bedrock topography datasets (Morlighem et al., 2014). If this trend continues and initialisation methods get further developed, it is reasonable to expect that the uncertainty in simulated ice sheet model evolution due to initialisation can be reduced for upcoming projections of the future.

The comparison shows that, despite all the differences, the ice sheet models that took part in this intercomparison agree well in their dynamic response to the SMB forcing for the region of overlap. This is an encouraging sign, given the large diversity of approaches. However, while this good agreement means that all models are able to accurately simulate changes in driving stress, other dynamic forcings (e.g., changes at the marine-terminating glaciers) were not included in the present set of experiments and may lead to a wider variety of responses. To achieve progress in this direction, we need a more complete understanding of the forcings and mechanisms that drive observed ice sheet changes. Aside from SMB, the important questions of how much surface melt water is reaching the bed, how the basal drainage system evolves and, most importantly, how the marine-terminating glaciers interact with the ocean in fjord systems are under active research.

The current 'ensemble of opportunity' approach, just as for GCMs, makes interpretation challenging: in other words, it is difficult to assess which choices in method and which uncertain model inputs have most influence on the results. Ideally, we would have liked to draw firmer conclusions about the influence of modelling choices on the quality of the initialisation and the uncertainty in modelled sea-level contribution. At the present stage, however, the sample size for a given modelling choice is often not sufficient and, more importantly, different model characteristics are not independent from each other. Similar difficulties have been discussed for the CMIP multi-model ensemble and may have led to the IPCC to resort to (slightly arbitrary) expert judgments for some interpretations. Improving the

uncertainty analysis and enabling a more rigorous intercomparison and evaluation would require an experimental design that is more controlled and prescriptive. Ice sheet models are well-placed to be used in such a design, being far less computationally expensive than e.g. GCMs, and with far fewer inputs to choose and outputs to evaluate. The effects of changing model structure (such as physics laws

and approximations, and resolution) on initialisations and projections is also far easier to evaluate. We therefore envision a second stage of the initMIP-Greenland experiments that performs multiple specific perturbations of the initial states of several models that can be interpreted in a statistically more meaningful way.

**Data availability**
The model output from the simulations described in this paper will initially be distributed via ftp server and at a later stage be included in the CMIP6 archive through the Earth System Grid Federation (ESGF) with digital object identifiers (DOIs) assigned. In order to document CMIP6's scientific impact and enable ongoing support of CMIP, users are obligated to acknowledge CMIP6, the participating

modelling groups, and the ESGF centres (see details on the CMIP Panel website at http://www.wcrp-climate.org/index.php/wgcm-cmip/about-cmip). The forcing datasets are equally initially available through the initMIP-Greenland wiki and in the future through the ESGF with version control and DOIs assigned.

**Acknowledgements**.
We acknowledge the Climate and Cryosphere (CliC) Project and the World Climate Research Programme (WCRP) for their guidance, support and sponsorship. We would like to thank Brice Noël for help with the SMB validation data and Xavier Fettweis, Miren Vizcaino and Vincent Cabot for providing SMB data.

Andy Aschwanden, Eric Larour, Sophie Nowicki and Helene Seroussi were supported by grants from NASA Cryospheric Science Program and Modeling Analysis and Prediction Programs. Heiko Goelzer has received funding from the program of the Netherlands Earth System Science Centre (NESSC), financially supported by the Dutch Ministry of Education, Culture and Science (OCW) under Grantnr. 024.002.001. Nicholas R. Golledge is supported by Royal Society of New Zealand Rutherford

Discovery Fellowship 15-VUW-004. Philippe Huybrechts acknowledges support from the iceMOD project funded by the Research Foundation-Flanders (FWO-Vlaanderen). William H. Lipscomb and Joseph H. Kennedy were supported by the Regional and Global Climate Modeling and Earth System Modeling programs of the U.S. Department of Energy's Office of Science. The National Center for Atmospheric Research is sponsored by the National Science Foundation. Victoria Lee, Stephen L.

Cornford and Antony J. Payne carried out work as part of the UKESM contribution for CMIP6. Mathieu Morlighem was supported by the National Aeronautics and Space Administration, Cryospheric Sciences Program (#NNX15AD55G), and the National Science Foundation's ARCSS program (#1504230). Christian Rodehacke (DMI) has received funding from the European Research Council under the European Community's Seventh Framework Programme (FP7/2007-2013) / ERC grant

agreement 610055 as part of the Ice2Ice project as well as the Nordic Center of Excellence eSTICC (eScience Tool for In-vestigating Climate Change in northern high latitudes) funded by Nordforsk (grant 57001). Florian A. Ziemen was supported by the BMBF project PALMOD. Computational

resources for MPI-PISM were made available by DKRZ through support from BMBF. IGE-Elmer simulations were performed using the Froggy platform of the CIMENT infrastructure, which is supported by the Rhône-Alpes region (GRANT CPER07_13 CIRA), the OSUG@2020 labex (reference ANR10 LABX56) and the Equip@Meso project (reference ANR-10-EQPX-29-01), and using HPC
resources from GENCI-CINES (Grant 2016-016066). Ralf Greve, Ayako Abe-Ouchi and Fuyuki Saito were supported by the Arctic Challenge for Sustainability (ArCS) project of the Japanese Ministry of Education, Culture, Sports, Science and Technology (MEXT), and by the Japan Society for the Promotion of Science (JSPS) KAKENHI under Grant Number 17H06104, Ralf Greve and Ayako Abe-Ouchi were additionally supported under JSPS Grant Number 16H02224 and Fuyuki Saito under JSPS
Grant Number 17K05664. Ayako Abe-Ouchi and Fuyuki Saito were supported by the Integrated Research Program for Advancing Climate Models (TOUGOU program) from MEXT. Reinhard Calov was funded by the Leibniz Association grant SAW-2014-PIK-1 and is now funded by the Bundesministerium für Bildung und Forschung (BMBF) grants PalMod-1.1-TP5 and PalMod-1.3-TP4.

## 6   Appendix A: SMB anomaly forcing

For the idealised forward experiment that serves to evaluate the initialisation, we have used a parameterisation of SMB anomalies (*dSMB*) as a function of surface elevation and latitude based on the following goals:

- to capture the first order pattern of the SMB changes that can be expected from the climate models that will be used in ISMIP6 projections
- to provide an idealized forcing, independent of one particular model or modelling choice
- to avoid masking problems by generating a forcing applicable to the whole model domain

The parameterisation has the form *dSMB* = f(sur, lat):

$$dSMB = \min[p_3 * (h - p_2) + p_4 * (\phi - \phi_0), p_1]$$

where *dSMB* is the SMB anomaly, *h* is the surface elevation, $\phi$ is the latitude and $\phi_0$ the reference
latitude in degrees. The parameters are the constant SMB anomaly in the accumulation area ($p_1$), the surface elevation of zero SMB anomaly ($p_2$), the gradient of SMB anomaly with elevation change ($p_3$), and the SMB anomaly change per degree latitude ($p_4$).

The target *dSMB* is calculated from differences in SMB between the periods 2080-99 AD and 1980-99 AD. We have fitted the parameters independently to output of three models of different complexity
(Table 4), one RCM (Fettweis et al., 2017), one general circulation model with elevation classes (GCM, Vizcaino et al., 2015) and one positive-degree-day model in combination with output from an Earth system model of intermediate complexity (EMIC-PDD, Goelzer et al., 2012).

**Table 4 Parameters with the best fit to the modelled data for SMB models of different complexity.**

| Parameter | $p_1$ (m yr$^{-1}$) | $p_2$ (m) | $p_3$ (m yr$^{-1}$ m$^{-1}$) | $p_4$ (m yr$^{-1}$ deg$^{-1}$) |
|---|---|---|---|---|
| RCM | 0.0720 | 2248.4 | 0.0016 | 0.1011 |
| GCM | 0.0549 | 2438.1 | 0.0007 | 0.0568 |
| EMIC-PDD | 0.0292 | 1642.1 | 0.0023 | 0.0462 |

The sensitivity of *dSMB* to elevation changes is around a factor 2 lower in the GCM compared to the RCM and is the highest in EMIC-PDD as can be seen by comparing $p_3$ in Table 4. We have used the parameter set of medium sensitivity (RCM) for the experiments.

Results for the RCM are shown in Figure A1 with the *dSMB* from the model (a) and from the parameterisation (b) in comparison. While the parameterisation allows for calculating *dSMB* everywhere on the grid, results are masked to the same extent as the modelled data, to facilitate comparison. These results show that the first order pattern is well captured by the parameterisation. The parameterisation works equally well for the two other climate models when proper masking is applied
to limit the calculation to ice covered regions (not shown).

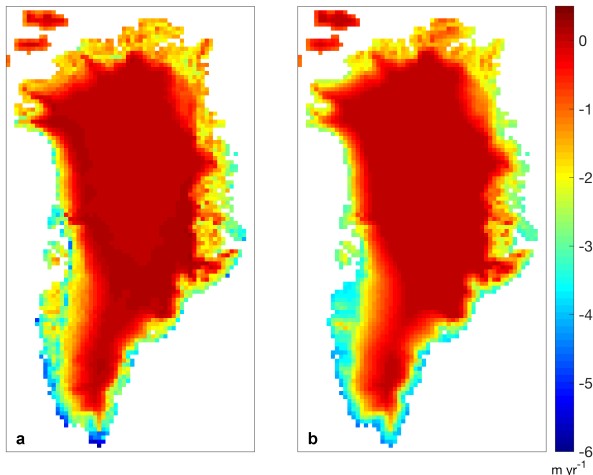

**Figure A1: SMB anomaly from a model (a) and reproduced by the parameterisation (b).**

For the final ISMIP6 forcing data, the parameterisation was applied to the observed geometry (Bamber et al., 2013) smoothed by a two-dimensional averaging filter (21x21 points). This step serves to produce a smooth forcing field for the range of expected model resolutions. The resulting dSMB on 1 km resolution was used to generate the forcing for other grids and resolutions by conservative interpolation.

For experiment *asmb*, the amplitude of the SMB anomaly was implemented as a time-dependent function, which increases stepwise every full year to 100 % at year 40. The amplitude is then held constant ($t > 40$ yrs) for prolongation of the experiment until year 100. The forcing is therefore independent of the time step in the individual models.

$$SMB(t) = SMB_{init} + dSMB * (floor(t)/40); \quad 0 \le t \le 40$$
$$SMB(t) = SMB_{init} + dSMB * 1.0; \quad\quad\quad t \ge 40$$

where $SMB_{init}$ is the SMB used for the initialisation in each individual model and *dSMB* is the provided SMB anomaly, which is identical for all models. The units of the *dSMB* in the provided data files are (meter ice equivalent/year) with an assumed density of 910 kg/m$^3$ and 31556926 s yr$^{-1}$. Note that for models assuming a different ice density, the input data have to converted accordingly.

## 7   Appendix B: Detailed model description

The models and initialisation methods of the individual models are documented in this section.

### 7.1      ARC-PISM

A similar approach to that used for previous Antarctic simulations is followed (e.g., Golledge et al.,
2014, 2015), but the length of the runs is modified on the basis that GrIS achieves thermal equilibrium
faster than e.g. the EAIS. Based on raw input data (Bamber et al., 2001) a 'shallow-ice' only run of 5
years is performed to reduce any spurious steep surface gradients in the data. From the output of this
run, a 50 kyr fixed-geometry run is performed, in which the ice sheet is allowed to come into thermal
equilibrium with the imposed (present-day) climate. The output from this run is then used for a 15,000
year spinup simulation, in which full model physics are employed, i.e. all model boundaries are allowed
to evolve. To minimize drift in this spinup run an initial exploration of parameter space is undertaken to
find an optimal combination of values. Parameter tuning is focused on 6 key controls: enhancement
factors for the SIA and SSA, maximum and minimum till friction angles, pseudo-plastic exponent 'q',
and the fraction of overburden pressure supported by the till. These parameters have been found to exert
the primary controls on location and magnitude of sliding and ice flow by deformation, and in doing so
most effectively control simulated ice-sheet geometry and volume. To identify an optimal
configuration, an initial ensemble of paired parameter simulations is performed, in which the variance
between each pair is assessed and all other variables are held constant. Simulations are run at 5 km
resolution for 500 years under unforced climatic conditions (i.e. present-day) but with freely-evolving
boundaries. Each run is assessed for degree of drift from initialisation. Subsetting from these
experiments, a further ensemble of 64 experiments is run, combining all combinations of 2 possible
values for each of the 6 parameters. The 'optimal' configuration is chosen based on 1) the lowest
deviation from present-day sea-level-equivalent ice volume and 2) the smallest domain-averaged
thickness mismatch at the end of the run compared to initialisation. For the latter metric, the standard
deviation of the mismatch was assessed, but differences between runs are minimal. These short runs
identify the relative control exerted by each parameter over 500 years. To achieve a much longer spinup
that deviates least from the starting conditions, a further seven experiments are undertaken until the
optimum parameter configuration is found. The final state of the spinup run is then used as the starting
point for the prognostic (*ctrl* and *asmb*) experiments.

### 7.2      AWI-ISSM

The thermo-mechanical coupled Ice Sheet System Model (ISSM, Larour et al., 2012) is used to create
an initial condition. For the initialization, a hybrid procedure that combines assimilation and a
temperature spin-up over longer periods is setup. The present-day ice sheet geometry (Morlighem et al.,
2016) is used and the observed horizontal surface velocities (Rignot and Mouginot., 2012) are
assimilated to infer the basal friction coefficient. After an initial relaxation of the ice sheet geometry for
years to avoid spurious noise (with no sliding and a constant temperature field), the temperature
spin-up is performed on a fixed topography with two different climatic forcings: AWI-ISSM1) present-
day climatic and AWI-ISSM2) paleo-climatic conditions. During the inversion, the ice viscosity is kept

constant using the enthalpy field from the end of the temperature spin-up. As the higher-order approximation to the Stokes flow is employed, grid refinements are made during the whole initialization procedure (grid sequence 1: hmin=15 km, hmax=50 km; grid sequence 2: hmin=5 km, hmax=50 km, grid sequence 3: hmin 2.5 km, hmax=35 km. In the vertical 17 layers refined to the base are used. AWI-ISSM1 is run for 20 kyr, 40 kyr and 5 kyr in each grid sequence while AWI-ISSM2 is run for 125 kyr, 125 kyr and 25 kyr). Geothermal flux, present day surface temperature and paleo surface temperature anomaly is taken from the SeaRISE webpage (http://websrv.cs.umt.edu/isis/index.php/Present_Day_Greenland). Surface mass balance is an annual mean for the period 1979-2014 from the downscaled RACMO2.3 model (Noël et al., 2016).

## 7.3     BGC-BISICLES

The initial state is found using data assimilation of velocity followed by relaxation of the surface elevation subject to a constant-in-time SMB (Lee et al, 2015). Merged surface ice velocity from Rignot and Mouginot (2012), is used to infer a 2-D basal traction coefficient and a 2-D stiffening factor multiplying the effective viscosity by solving an inverse problem with fixed ice sheet geometry from Morlighem et al. (2014). The ice surface is evolved by forcing the model using the 2-D parameters with a 1997-2006 mean SMB from HIRHAM5 (Lucas-Picher et al., 2012), subject to fixed calving front boundary conditions. The surface is relaxed in this way for 120 years, which is sufficient for the absolute value of the instantaneous rate of change ice thickness to fall below 0.5 m a-1 in 99 percent of the total area of GrIS. This initialization uses a 3-D, steady-state temperature field generated by a high-order thermomechanical model by Price et al., 2011. For the *ctrl* and *asmb* experiments the fixed calving front is replaced by a calving model (Taylor 2016), where ice calves if water-filled, surface crevasses reach a depth equal to the height of the ice above sea level. A basal melt rate varying between 0 and 4 times ice thickness is also applied in regions where ice is close to fracture.

## 7.4     DMI-PISM

Spinup over one full glacial cycle (125 kyr BP to present) with the following guidelines: freely evolving run that inherits the climate memory of the last glacial-interglacial cycle and shall represent the currently observed ice sheet state for the contemporary "year of assignment". Since we at DMI focus on coupled climate model-ice sheet model simulations, we value a free run that is consistent with the applied forcing higher than a perfect representation of the current observed Greenlandic ice sheet state, such as ice sheet geometry. We have found that this procedure is necessary to avoid strong unnatural drifts in the ice sheet model component after the full coupling between climate model and the ice sheet model is established (Svendsen et al., 2015). The spinup first goes through one complete glacial-interglacial cycle using as base the Era-Interim reanalysis of the period 1979-2012 to determine the surface mass balance (SMB) via positive degree days (PDD). The scaling of the datasets is determined based on the Greenland temperature index in the SeaRISE Greenland Dataset (based on ice core data; source SeaRISE Reference data set: Greenland_SeaRISE_dev1.2.nc). Temporal evolution of the sea level is also taken from the same SeaRISE Greenland dataset. The ensemble of runs (PISM1, PISM2, PISM3, PISM4, PISM5) differ in the forcing applied to the Greenland Ice Sheet (GIS). In all cases the forcing source is based on the Era-Interim reanalysis covering the period 1979-2012. The only

differences are the applied PDD factors for the determination of the surface mass balance (SMB) via positive degree days (PDD). The following enumeration lists the applied different PDD factors. PISM0: PDD_snow=0.012 m/°C day, PDD_ice=0.018 m/°C day; PISM1: PDD_snow=0.010 m/°C day, PDD_ice=0.016 m/°C day; PISM2: PDD_snow=0.009 m/°C day, PDD_ice=0.014 m/°C day; PISM3:

PDD_snow=0.008 m/°C day, PDD_ice=0.012 m/°C day; PISM4 PDD_snow=0.004 m/°C day, PDD_ice=0.008 m/°C day.

## 7.5    IGE-ELMER

The model is initialised using a control inverse method as in Gillet-Chaulet et al. (2012). For the momentum equations, we solve the Shelfy-Stream Approximation (SSA). The vertically-averaged

viscosity is constant in all simulations and is initialised using the temperature field coming from a paleo spin-up (125ky) of the SICOPOLIS model. The limit of the model domain is fixed and corresponds to the boundary with the ocean: calving front positions are fixed and the calving rate is computed as the opposite of the ice flux through the boundary; land-terminated parts can freely retreat or advance up to the domain limit. The ice-sheet topography is initialised using the IceBridge BedMachine Greenland V1

dataset (Morlighem et al., 2014) where missing values for the bathymetry around Greenland have been filled using data from Bamber et al. (2013). We use a linear basal friction law. The basal friction coefficient is constant in all transient simulations and is initialised with the control method so that the mismatch between observed and modelled velocities is minimum. As observations, we use a composite from the MEaSUREs Greenland Ice Sheet Velocity Map (V1) (Joughin et al., 2010). The ice sheet

model is then relaxed for 20 years using a 1989-2008 mean SMB from the regional climate MAR forced with ERA-Interim. The only difference between IGE-ELMER1 and IGE-ELMER2 is the mesh resolution as given in Table 3.

## 7.6    ILTS-SICOPOLIS

The model is SICOPOLIS version 3.3-dev in SIA mode and with the melting-CTS enthalpy method

(ENTM) for ice thermodynamics by Greve and Blatter (2016). The present-day surface temperature parameterisation is by Fausto et al. (2009), the present-day precipitation is by Ettema et al. (2009) and the geothermal heat flux is by Greve and Herzfeld (2013) (slightly modified version of the heat flux map by Greve (2005)). A spin-up over the last glacial-interglacial period (125,000 years) is carried out. Except for initial and final 100-year phases with freely evolving surface and bedrock topography, the

topography is kept fixed during the spin-up, whereas the temperature evolves freely. This is essentially the method that was used for the SeaRISE experiments (documented in detail by Greve and Herzfeld, 2013). The time-dependent forcing for the spin-up is the GRIP $\delta^{18}$O record (Dansgaard et al., 1993; Johnsen et al., 1997) converted to a purely time-dependent surface temperature anomaly $\Delta T$ by the conversion factor 2.4°C/‰ (Huybrechts, 2002).

## 35  7.7    ILTSPIK-SICOPOLIS

The model version, thermodynamics solver and present-day surface temperature parameterisation are the same as listed in Section 7.6. The present-day precipitation is by Robinson et al. (2010) and the

geothermal heat flux is produced by Purucker (https://core2.gsfc.nasa.gov/research/purucker/heatflux_updates.html) following the technique described in Fox Maule et al. (2005). The ice discharge parameterisation by Calov et al. (2015), Eq. (3) therein with the discharge parameter c = 370 m$^3$ s$^{-1}$, is applied. A spin-up over the last glacial-
interglacial period (135,000 years) with free evolution of all fields (including the ice sheet topography) is carried out. The time-dependent forcing for the spin-up is the GRIP $\delta^{18}$O record (Dansgaard et al., 1993; Johnsen et al., 1997) on the GICC05 time scale (Svensson et al., 2008), converted to a purely time-dependent surface temperature anomaly $\Delta T$ by the conversion factor 2.4°C/‰, and further a 7.3% gain of the precipitation rate for every 1°C increase of $\Delta T$ (Huybrechts, 2002).

**7.8    IMAU-IMAUICE**

The model (de Boer et al., 2014) is initialised to a thermo-dynamically coupled steady state with constant, present-day boundary conditions for 200 kyr using the average 1960-1990 surface temperature and SMB from RACMO2.3 (van Angelen et al., 2014), extended to outside of the observed ice sheet mask using the SMB gradient method (Helsen et al., 2012). Bedrock data is from Bamber et al. (2013)
and geothermal heat flux from Shapiro and Ritzwoller (2004). The model is run in SIA mode with ice sheet margins evolving freely within the observed coast mask, outside of which ice thickness is set to zero.

**7.9    JPL-ISSM**

The ice sheet configuration is set up using data assimilation of present-day conditions and historical
spin-up similar to the study of Schlegel et al. (2013). SSA is used over the entire domain with a resolution varying between 1 km in the fast-flowing areas and along the coast and 15 km in the interior. Grounding line migration is based on hydrostatic equilibrium and a sub-element scheme (Seroussi et al., 2014). Observed surface velocities (Rignot and Mouginot, 2012) are first used to infer unknown basal friction at the base of the ice sheet (Morlighem et al., 2010). Ice temperature is modeled assuming the
ice sheet to be in a steady-state thermal equilibrium (Seroussi et al., 2013). A spin up of 50,000 years is then done to relax the ice sheet model (Larour et al., 2012) and reduce the initial unphysical transient behavior due to errors and biases in the datasets (Schlegel et al., 2016) using mean surface mass balance from 1979-1988 (Box, 2013). A historical spin up is then done from 1840 to 2012 using reconstructions of surface mass balance for this period (Box, 2013). Bedrock topography is interpolated from the
BedMachine dataset (Morlighem et al., 2014), that combines a mass conservation algorithm for the fast-flowing ice streams and krigging in the interior of the ice sheet. Initial ice thickness is from the GIMP dataset (Howat et al., 2014). Geothermal flux is from Shapiro and Ritzwoller (2004), air temperature from RACMO2 (van Angelen et al., 2014). SMB from a mass balance reconstruction (Box, 2013) averaged over the 2000-2012 period is used in the *ctrl* experiment.

**7.10    LANL-CISM**

The ice sheet was initialised with present-day geometry and an idealized temperature profile, then spun up for 20,000 years using pre-1990 climatological surface mass balance and surface air temperature

from RACMO2. No glacial data were used. The model was spun up for 20,000 years to equilibrate the temperature and geometry with the forcing. The model was initialised (prior to spin-up) with present-day topography and thickness based on the mass-conserving bed method of Morlighem et al. (2011). The surface mass balance (SMB) over the ice sheet was a 1961-1990 climatology from RACMO2. In grid cells where RACMO2 did not provide an SMB, the SMB was set arbitrarily to -2 m yr$^{-1}$. Surface air temperatures were also from a 20th century RACMO2 climatology (Ettema et al. 2009). The geothermal flux was set spatially uniform to 0.05 W/m2.

### 7.11 LSCE-GRISLI

The GRISLI spin-up procedure is based on an iterative data assimilation method to infer the basal drag from the observed surface velocities. The first step consists in a 30 kyr equilibrium simulation of the internal temperature with prescribed ice sheet topography (Bamber et al., 2013), 1979-2005 averaged near surface air temperature (Fettweis et al., 2013), geothermal heat flux (Fox Maule et al. 2009), surface velocities (Joughin at al., 2013) and spatially variating basal drag coefficient from a previous GRISLI experiment (Edwards et al., 2014b). From the resulting internal fields, the 1979-2005 mean SMB and near-surface air temperature (Fettweis et al., 2013) is used to run a succession of eight 220-yr simulations. During the first 20 years, the basal drag coefficient is corrected to limit the deviation from prescribed velocities, and then the basal drag is kept constant for 200 years of surface relaxation. At each iteration, we update the basal drag coefficient with the value computed at the previous iteration. The prescribed velocities are the observed velocities corrected for thickness differences at the end of the 220 years in order to keep the ice flux in GRISLI identical to the observed one. Then, a second temperature equilibrium is run for consistency between the temperature field and the inferred basal drag coefficient. From this, an additional 220-yr simulation is run to optimise the final basal drag coefficient. This basal drag coefficient and associated final ice-sheet conditions are used as initial conditions for all the initMIP GRISLI experiments.

### 7.12 MIROC-ICIES1

The simulation set-up of MIROC-IcIES1 is described in Appendix A of Saito et al (2016), as the experiment E"s:e1:vm. The surface mass balance field to force the ice-sheet model follows the present-day field provided by SeaRISE, without any correction except for the horizontal resolution. This mass balance is computed using a PDD method, and the parameters are described at http://websrv.cs.umt.edu/isis/index.php/Future_Climate_Data. The field of basal sliding coefficients are relaxed such that the simulated ice-sheet topography under the present-day surface mass balance field mostly matches the observed geometry, using the method by Pollard and DeConto (2012b). Using the deduced basal sliding coefficients field, a steady-state spin-up under present-day climate condition with the fixed geometry is performed again while the temperature evolves freely.

### 7.13 MIROC-ICIES2

The simulation set-up of MIROC-IcIES2 is described in Saito et al (2016), as the experiment B':v2. A free spin-up over 125,000 years is performed following the SeaRISE configuration: the background

temperature history based on the oxygen isotope record of the GRIP ice core is used as anomaly to the present-day field. During the spin-up, the ice-sheet margin is allowed to freely advance and retreat. The present-day surface temperature follows the parameterisation presented in Fausto et al. (2009). The present-day mean annual precipitation follows Ettema et al. (2009). The surface mass balance is
computed using these fields and a PDD method whose coefficients follow those in Huybrechts and de Wolde (1999). The basal-sliding velocity is computed using the Weertman sliding law, with an allowance for sub-melt sliding following Hindmarsh and Le Meur (2001). The parameters are kept constant and follow those in Huybrechts and de Wolde (1999) except that the coefficient is doubled to obtain better match with the present-day topography.

**7.14    MPIM-PISM**

Spinup over one full glacial cycle (135 kyr BP to present), changed parameters at 20 kyr BP (simply faster starting from a pre-spun up state at 20 kyr BP than re-running the full glacial cycle for each param change). The spinup first goes through one complete glacial cycle using a linear combination of MPI-ESM output. The scaling of the two datasets is determined based on the Greenland temperature index in
the SeaRISE Greenland Dataset (based on GRIP data). Sea level changes are also taken from the SeaRISE Greenland Dataset.

**7.15    UAF-PISM**

Spin-up over a glacial cycle combined with a short relaxation run. To define the energy state, a "standard" glacial cycle run is performed where the surface can evolve freely, similar to Aschwanden et
al. (2013) and Aschwanden et al. (2016). The spin-up starts at 125 kyr BP with the present-day topography from Howat et al. (2014) using a horizontal grid resolution of 9 km. The grid is refined to 6 km, 4.5 km, and 3 km at 25 kyr BP, 20 kyr BP and 15 kyr BP, respectively. We use a positive degree-day scheme to compute the climatic mass balance from surface temperature (Fausto et al., 2009) and model-constrained precipitation (Ettema et al., 2009). The degree-day factors are the same as in
Huybrechts (1999). Second, we account for paleo-climatic variations by applying a scalar anomaly term derived from the GRIP ice core oxygen isotope record (Dansgaard et al., 1993) to the temperature field (Huybrechts, 2002). Then we adjust mean annual precipitation in proportion to the mean annual air temperature change (Huybrechts, 2002). Finally, sea level forcing, which determines the land area available for glaciation, is derived from the SPECMAP marine $\delta^{18}$O record (Imbrie et al., 1992). At the
end of the spin-up, the computed surface elevation differs from the observed surface elevation. From here we perform two sets of 60-year relaxation simulations using the RACMO 1960-1990 averaged climatic mass balance. In one set (UAF-PISM4-6), we regrid the spun-up state from the 3-km simulation to 1.5 km (UAF-PISM4), 3 km (UAF-PISM5) and 4.5 km (UAF-PISM6) and run a relaxation where the ice sheet is free to evolve. At the end of this relatively short relaxation, the
computed surface elevation continues to differ substantially from present-day observation and the model states exhibit a large artificial drift. To reduce the mismatch between observed and simulated surface elevations, we perform a second set, UAF-PISM1-3. Here we regrid the energy state in the ice and in the bedrock from the spun-up state from the 3-km simulation to 1.5 km (UAF-PISM1), 3 km

(UAF-PISM2) and 4.5 km (UAF-PISM3) and combine those fields with the present-day topography from Howat et al. (2014) to again run a relaxation where the ice sheet is free to evolve.

### 7.16    UCIJPL-ISSM

The ice sheet configuration is set up using data assimilation of present-day conditions (Morlighem et al., 2010). A relaxation of 50 years is then performed to reduce the initial unphysical transient behavior due to errors and biases in the datasets (Seroussi et al., 2011), using mean surface mass balance from 1961-1990 (van Angelen et al., 2014). A Higher-Order model (HO) is used for the entire domain, with 14 vertical layers and a horizontal resolution varying between 0.5 km along the coast and 30 km inland. We perform the inversion of basal friction assuming that the ice is in thermomechanical steady state. The ice temperature is updated as the basal friction changes and the ice viscosity is changed accordingly. At the end of the inversion, basal friction, ice temperature and stresses are all consistent. After the data assimilation, the model is relaxed for 50 years using the mean surface mass balance of 1961-1990 from RACMO (van Angelen et al., 2014), while keeping the temperature constant. Bed topography is interpolated from the BedMachine Greenland v3 dataset (Morlighem et al., 2017), that combines a mass conservation algorithm for the fast-flowing ice streams and kriging in the interior of the ice sheet. Initial ice surface topography is from the GIMP dataset (Howat et al., 2014). For the thermal model, surface temperatures from Fausto et al. (2009) and geothermal heat flux from Shapiro and Ritzwoller (2004) are used. Mean surface mass balance of 1961-1990 from RACMO (van Angelen et al., 2014) is used in the *ctrl* experiment.

### 7.17    ULB-FETISH

Model initialisation is based on the method by Pollard and DeConto (2012b) by optimizing basal sliding coefficients for the grounded ice sheet in an iterative way through minimizing the misfit between observed and modelled surface topography. A regularization term is introduced to smooth high-frequency noise in the basal sliding coefficients (Pattyn, 2017). Initial ice sheet surface and bedrock elevation are taken from Bamber et al. (2013) and geothermal heat flux stems from Fox Maule et al. (2009). The initialisation runs over a period of 50,000 years forced by a constant surface mass balance (Fettweis et al., 2007) and surface temperature (Fausto et al., 2009). During this time, the marine boundaries are kept fixed in space. For the control and forcing runs, the grounded ice margin and grounding line is allowed to move freely, starting from the initialized state. Two model setups were considered: FETISH1 is according to SIA; FETISH2 is a hybrid model (superimposed SSA-SIA) with a flux condition at the grounding line according to Schoof (2007) and Pollard and DeConto (2012a).

### 7.18    VUB-GISM

The model is initialised with a glacial spin-up over the last two glacial cycles and recent climate forcing data up to the year 2005 (Fürst et al., 2015). For the spin-up, a synthesized temperature record is used based on ice-core data from Dome C, NGRIP, GRIP, and GISP2 (Barker et al., 2011; Andersen et al., 2004; Dansgaard et al., 1993; Kobashi et al., 2011) and precipitation is scaled by 5% per °C. For the period 1958 to 2005, the atmospheric forcing comes from a combination of ECMWF ERA-

meteorological reanalysis and ECMWF operational analysis data. Use is made of monthly temperature anomalies and yearly precipitation ratios. The ocean forcing from 1958 to 2005 derives from a CMIP5 model providing temperature anomalies at mid-depth (300-800 m) in 5 surrounding ocean basins with respect to the 1960-1990 period. After the year 2005, atmospheric and oceanic forcings are reset to their 1960-1990 averages in the unforced state. Bedrock elevation and coast mask are based on Bamber et al. (2013), the pattern of surface accumulation for the period 1950-2000 is based on Bales et al. (2009). The higher-order model (GISM1) is initialised with an SIA model (GISM2) to 3 kyr BP. Switching at 3 kyr BP appeared to be sufficiently early to resolve the main effects of including horizontal stress gradients by the present day.

## 8    Appendix C: Data request

The requested variables (Table C1) serve to evaluate and compare the different models and initialisation techniques.

All 2D data were requested on a regular grid with the following description: Polar stereographic projection with standard parallel at 71° N and a central meridian of 39° W (321° E) on datum WGS84. The lower left corner is at (-800000 m, -3400000 m) and the upper right at (700000 m, -600000 m). This is the same grid (Bamber et al., 2001) as used to provide the SMB anomaly forcing. The output was submitted on a resolution adapted to the resolution of the model and was 10 km or 5 km. The data were conservatively interpolated to 5 km resolution for diagnostic processing.

If interpolation was required in order to transform the SMB forcing (1 km, same as Bamber et al., 2013) to the native model grid, and transform model output to the initMIP output grid (20 km, 10 km, 5 km, 1 km, Bamber et al., 2001), it was requested that conservative interpolation was used. The motivation for using a common method for all models is to minimize model-to-model differences due to the choice of interpolation methods. In most cases this has been followed by the modellers.

We distinguish between state variables (e.g., ice thickness, temperatures and velocities) and flux variables (e.g., SMB). State variables were requested as snapshot information at the end of one-year (scalars) and five-year periods (2D), while flux variables were averaged over the respective periods. For calculation of scalar diagnostics (e.g., total ice mass or ice covered area), it is necessary to correct for the area distortions implicit for a given projection (e.g., Snyder, 1987). Some of the variables may not be applicable for each model, in which case they were omitted.

**Table C1** Data request for participation in initMIP-Greenland.

Type: FL= Flux variable, ST= State variable, CST= Constant

| Variable name | Units | Type | Standard Name (CF) |
|---|---|---|---|
| Ice Sheet Altitude | m | ST | surface_altitude |

| | | | |
|---|---|---|---|
| Ice Sheet Thickness | m | ST | land_ice_thickness |
| Bedrock Altitude | m | ST | bedrock_altitude |
| Bedrock Geothermal Heat Flux | W m$^{-2}$ | CST | upward_geothermal_heat_flux_at_ground_level |
| Surface mass balance flux | kg m$^{-2}$ s$^{-1}$ | FL | land_ice_surface_specific_mass_balance_flux |
| Basal mass balance flux | kg m$^{-2}$ s$^{-1}$ | FL | land_ice_basal_specific_mass_balance_flux |
| Land ice calving flux | kg m$^{-2}$ s$^{-1}$ | FL | land_ice_specific_mass_flux_due_to_calving |
| Ice thickness imbalance | m s$^{-1}$ | FL | tendency_of_land_ice_thickness |
| X-component of land ice surface velocity | m s$^{-1}$ | ST | land_ice_surface_x_velocity |
| Y-component of land ice surface velocity | m s$^{-1}$ | ST | land_ice_ surface_y_velocity |
| Z-component of land ice surface velocity | m s$^{-1}$ | ST | land_ice_ surface_upward_velocity |
| X-component of land ice basal velocity | m s$^{-1}$ | ST | land_ice_basal_x_velocity |
| Y-component of land ice basal velocity | m s$^{-1}$ | ST | land_ice_basal_y_velocity |
| Z-component of land ice basal velocity | m s$^{-1}$ | ST | land_ice_basal_upward_velocity |
| X-component of land ice vertical mean velocity | m s$^{-1}$ | ST | land_ice_vertical_mean_x_velocity |
| Y-component of land ice vertical mean velocity | m s$^{-1}$ | ST | land_ice_vertical_mean_y_velocity |
| Surface Temperature of Ice Sheet | K | ST | temperature_at_ground_level_in_snow_or_firn |
| Basal Temperature of Ice Sheet | K | ST | land_ice_basal_temperature |
| Basal drag | Pa | ST | magnitude_of_land_ice_basal_drag |
| Land ice area fraction | 1 | ST | land_ice_area_fraction |
| Grounded ice area fraction | 1 | ST | grounded_ice_sheet_area_fraction |
| Floating ice sheet area fraction | 1 | ST | floating_ice_sheet_area_fraction |
| Ice Mass | kg | ST | land_ice_mass |
| Ice Mass not displacing sea water | kg | ST | land_ice_mass_not_displacing_sea_water |
| Area covered by grounded ice | m$^{2}$ | ST | grounded_land_ice_area |
| Area covered by floating ice | m$^{2}$ | ST | floating_ice_shelf_area |

| Total SMB flux | kg s$^{-1}$ | FL | tendency_of_land_ice_mass_due_to_surface_mass_balance |
|---|---|---|---|
| Total BMB flux | kg s$^{-1}$ | FL | tendency_of_land_ice_mass_due_to_basal_mass_balance |
| Total calving flux | kg s$^{-1}$ | FL | tendency_of_land_ice_mass_due_to_calving |

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
