# Peer review of "Design and results of the ice sheet model initialisation experiments initMIP-Greenland: an ISMIP6 intercomparison"

_The Cryosphere, 2017_

## Referee Comment (RC1) · Anonymous Referee #1 · 16 Aug 2017

**Summary**

This paper summarizes models, methods, and the results of ice sheet model initialization experiments for the Greenland ice sheet, which are (eventually) targeting the ISMIP6 contribution to CMIP6. The broad motivation for the exercise is the recognition – largely motivated by previous, community-endorsed ice sheet model intercomparisons – that decadal and century scale simulations conducted using present-day ice sheet models are keenly sensitive to the model initial conditions (so much so that transients in the initial condition often swamp the ice sheet model response to the prescribed climate forcing of interest). In this paper, the different initialization methods most com-

monly used by the ice sheet modeling community are discussed, as are the perceived pros and cons of the different approaches. A set of forward model experiments are conducted, which are intended to clearly expose the magnitude and trend of (1) model initialization transients under steady forcing and (2) model transients under idealized climate forcing, here applied in the form of surface mass balance (SMB) anomalies.

While the results of the simulations and paper are not necessarily in themselves scientifically compelling, they serve as an important record and waypoint for the purpose of documenting the current practices and capabilities of different modeling groups, as well as the differences in standard model outputs when applying these different models to the same set of experiments. Therefor, while this paper does not necessarily represent any major scientific breakthroughs, I think it is worthy of publication in TC purely for the purpose of clearly documenting where the ice sheet modeling community stands on these issues at this time. The paper represents the combined and significant efforts of a large number of independent ice sheet modeling groups, including the countless hours spent on model development and testing prior to applying the models to any experiments.

I disagree a bit with the way some of the conclusions for the experiments have been presented or "pitched" here. For one, it's not clear to me that the spread and/or size of the transients in the steady-forcing initialization experiment do actual show improvement over previous efforts. This could be due, in part, to the way the data are presented (discussed further below). It could also be that a simple quantitative metric from this study, relative to previous studies, needs to be included to support this statement. Further, the fact that the perturbation experiment uses anomalies applied on top of the SMB used for initialization is already an indication that we aren't looking at an apples-to-apples comparison of response between models. For example, if an actual SMB forcing (as opposed to a set of anomalies) was prescribed for the perturbation experiment, the spread in sea-level-equivalent mass loss shown in Fig. 8c would likely be much larger. I don't necessarily disagree with this choice, but the summary statements

don't seem entirely accurate.

One obvious place for improvement is in the conclusions, which currently do not provide any clear guidance as to a "path forward", in the sense of what did we learn here and where should the community be working towards in order to address some of the issues and problems discussed in this paper. It seems like one obvious conclusion that could be stated more clearly is that either using data-assimilation OR spin-up approaches in isolation is not a long-term solution. Rather, using some combination of these two approaches is going to be necessary if the goal is to initialize models to have both a realistic initial state while simultaneously minimizing unrealistic transients (good) or representing actual observed ice sheet transients (better). There are some references (noted below) that could be included for these types of approaches, which aren't currently being applied at the whole-ice-sheet scale, but which should be viable for doing so in the near future. This would help to end the paper on a more positive, forward-looking note.

Below, comments are tied to specific points in the submitted text using the key "X, Y", where X refers to the page number and Y refers to the line number.

**Major Comments**

1,18: I'm not sure I really agree with the last sentence. If the authors really think this is true, then they should back it up quantitatively somewhere in the text. Is the drift referred to here for the init. and steady-forcing experiment? The SMB anomaly experiment? Both?

2,3: "encapsulates most of the modeling decisions . . ." – I don't entirely agree with this. A major "modeling decision" would be whether or not I choose to use SIA or a Stokes model, but I can imagine using the same init. procedure for both if I chose to do so. It might make more sense here to just point out that the decisions made during the init. process have a very strong impact on any model outputs that follow from using that initial state. I think the point you are trying to make here is simply that these choices

are critical as they have very large consequences for "projections".

2,28-30: The last sentence here is not really true. For example, there are fairly extensive "internal" constraints available in the form of radar layers, dated at ice core sites and tracked over hundreds of km of the ice sheet. Even though we are not regularly using these data now as constraints, we should be, or could be in the future. Maybe this is a point to make in the conclusions / future directions part of the paper, with appropriate references to some of the work by Joe MacGregor (which is particularly relevant to Greenland)?

3,28-30: "All three methods . . .". Note that these could / can all be combined, and this is probably the future direction we as a community should be moving towards. It might be worth noting that here, even though none of the currently participating models do this (and then discuss this again in the conclusions by pointing to a few papers that are exploring what would truly be called data assimilation – that is, trying to assimilate observations of change (like dh/dt) and/or fields that include a record of change over time (like temperature, layer shapes)).

3,7-33: In general, I think the major problem being discussed here could be stated a bit more succinctly. That is, the spin-up approach provides an initial condition that has a realistic internal TRANSIENT with respect to some applied external climate forcing, but also generally gives us an initial STATE that is a poor representation of the present day. The current incarnation of optimization approaches does the opposite: we get a good representation of the present-day ice sheet state, but a very bad representation of the internal transients. The solution going forward should be to formally combine these two seemingly at-odds approaches through improved data assimilation approaches.

5,2-3: It is probably important to be explicit about the "lineage" of models here. While there are 17 group contributions, more than half of these are from "repeated" models (i.e., 9 of the models listed are either ISSM, PISM, or SICOPOLIS), so in effect, there are only really 11 different models represented here.

6,2-23: It might be nice to point out that the division of the three initialization strategies is a bit arbitrary, and not perfect. For example, most of the models that use data assimilation also use some form of spin-up (e.g., for ice temperatures). Also, the "DAs" method is really a sort of "ad hoc" approach to assimilation – instead of using formal methods to iteratively minimize a PDE-constrained, observation-minus-model cost functional, these approaches are minimizing the observation-minus-model difference using some other method (i.e., adjusting the basal traction locally to minimize the mismatch with surface elevations). I appreciate that some categorization needs to be made here to bin these approaches, but it's worth pointing out that the categorization is far from perfect. It's actually quite fuzzy and this is because it's starting to be obvious that some combination of these different methods is required.

10,5-21: It's not entirely clear to me what the purpose of this section is. It's missing a clearly motivating intro / closing sentence to make that clear.

Figure 8: If you are going to keep panel (a) of this figure as is, then I think you also need to show a 2nd figure where you normalize by the initial volume (in SLE). As shown, these trends all look very reasonable but this is because you are showing the rate of absolute, rather than relative change. When normalized by the initial volumes, we would more easily see the volume changes with the more relevant units of cm rather than m. In panels (b) and (d), what time periods are the trends calculated over? In panel (c), the vertical axis label is "mass change". For this, a positive sign would suggest ice sheet growth. Should the wording be changed (e.g., "mass loss") so that it is clear we are looking at mass loss (which is obvious if we are talking about SLR)?

16,9: "inconsistencies between ice velocity and geometry datasets". The reference to Seroussi et al. won't help the unfamiliar reader here. I suggest you just be explicit about what the problem is: the optimized model velocities, while possibly being consistent with observed surface velocities, conspire with thickness errors to give a flux divergence that is wildly different than and unbalanced by the local SMB. This is where the large and unrealistic thickness transients come from when conducting a forward

run following initialization using a velocity-based-only optimization approach. Note that the uppermost panel in Figure 10 shows this nicely – the very noisy, large amplitude thickness adjustments in the ISSM model are a manifestation of this (relative to the much more subtle thickness transients for the spun-up ICIES2 and PISM models).

19,16-18: This is left hanging a bit. As noted above in the summary comments, instead of simply pointing out that none of the methods discussed here prove to be optimal for initialization, you could take the opportunity to discuss recent efforts that target these deficiencies (even if they are not yet recognizable at the full-ice sheet scale) and push for their exploration, development, and wider adoption. I'm thinking in particular about formal efforts that attempt to include additional constraints and / or assimilate time dependent observations (for example, Goldberg et al., TC, 7(6), 2013; Perego et al., J. Geophys. Res., 119, 2014).

19,20: "DAv is the method of choice for short-term projections." I strongly disagree with this statement. Unless you are talking about applying anomaly forcing (as in the perturbation experiment discussed here), this is absolutely not true. Most often, these are going to have the worst and most non-physical thickness transients when stepped forward in time under realistic climate forcing. Only if that climate forcing has been taken into consideration during the optimization process do these have a hope of minimizing non-physical transients. This can clearly be seen in the top left figure of Figure 10 (the ISSM, DAv initialized model has a noisy, large amplitude thickness transient that is very unphysical). In general, this paragraph is not true unless you are talking about anomaly forcing experiments, which I think in general, the community is trying to move away from (for example, an initial condition generated to work well only when applied in an anomaly forcing experiment is of little use for coupling to a climate model, since climate models are generally required to conserve mass, and anomaly forcing does not do that).

20,7-9: I am not yet convinced that this statement – here and in the abstract – is supported by the results shown in this paper. It could be the case, but based on Figure

8a, it is very hard to judge. We would have to see Figure 8a normalized by initial ice sheet volume to be convinced of this. Also, if this statement is going to be supported, it would be nice to do so quantitatively based on previous intercomparison results (e.g. compare the spread from a similar experiment from SeaRISE with that from this study).

20,11-13: "If this trend continues . . .". While I agree with this statement, I don't think it is for the same reasons as discussed in this paper. If there's any reason this trend will continue to improve, it's that better formal optimization and initialization methods are being developed and applied to ice sheets. That is, methods that take into account not only observational constraints on the current ice sheet state, but also observational constraint on short (i.e. dh/dt) and longer (i.e., temperature profiles) term trends that are inherent to the ice sheet. Again, I feel like there is a bit of a missed opportunity in the conclusions of this paper to point out and promote some of these new directions and methods that should effectively minimize many of the problems highlighted and discussed in this paper.

**Minor / Editorial Comments**

Title: Check that the spelling of "initialisation" is correct? Autocorrect seems to prefer "initialization".

1,2-3: Should summary refs. for the SeaRISE and Ice2Sea projects be included here? Similarly, should ISMIP6 and CMIP6 efforts here point to some generic reference publications?

1,6: ". . . to estimate the associated uncertainties." (in what? Model outputs?)

1,10: schematic -> idealized ?

1,13-14: Is there really a wide diversity in the data sets and boundary conditions used? It seems like for the most part, the models are using similar datasets and boundary conditions (for the latter, I mean w.r.t. "observation-based" boundary conditions – I understand that things like basal boundary conditions, which include assumptions about

sliding, thermal state, hydrology, etc., probably DO vary widely).

1,21-28: The initial reference to EISMINT, followed by a listing of many other non-EISMINT intercomparisons, is confusing. I understand that you only mean to use the categorization that EISMINT suggested, but it could be read as if everything underneath (e.g., ISMIP-HOM, MISMIP, etc.) was also "part of" EISMINT. Suggest rewording this slightly?

1, general: Do CMIP and ISMIP6 need to be defined here?

2,10: "removing mass at the margins" – this is a little unclear, as SMB also removes mass inland (although maybe this mass is transferred to the margins, e.g. as melt).

2,32-34: I would be more emphatic about this point – without very special care, the transient from initialization can entirely dominate the response in decadal / century scale projections.

2,41: When you say run forward in time here, it might help to add detail about the timescale you are thinking about (e.g., thousands to tens of thousands (more?) years, rather than decades or centuries).

3, 2: "the model's state is internally consistent" – It isn't clear what you mean by this. The model's internal state is always internally consistent, regardless of the type of spin-up you do. I think what you mean is that models that go through a spin-up are, at any point in time, closer to being in equilibrium with the applied climate forcing, or at least contain internal transients (e.g., in temperature) that are more consistent with external forcing transients. I appreciate this is a subtle difference to try to express.

3,10: "by inversion" – be a bit more specific here. I think you mean formal, PDE-constrained optimization?

3,16-18: Other "errors" that are forced to be accounted for by optimized parameter fields include observations and model state variables that are entirely ignored in the optimization process. For example, one of the biggest reasons current optimization

approaches lead to such large, unphysical transients is that they are ignoring whether or not the temperature (and hence rheology) is consistent with observations (and as such, all of that uncertainty gets pushed into the basal parameter optimization).

3,19-22: Here, I think you need to be explicit in noting that the thing that is being adjusted is the bedrock elevation (i.e., we have good constraints on the ice sheet surface elevation, but much less so with the bedrock elevations. So the thickness is adjusted with the assumption that the uncertainty is the bedrock location, not the upper surface location).

4,5: "to a large perturbation" – add "in SMB forcing"? or "in climate forcing"?

4,17: "... that all boundary conditions AND FORCING remain constant in time." ?

Table 1: "schematic change of SMB forcing" -> "idealized change in SMB forcing"

Table 2: Is the list of "contributors" consistent with co-authors on the paper, or is it supposed to be more of a list of co-authors for the respective modeling efforts?

6,12-14: Note that SP approaches are generally also favored when the goal is to include an ice sheet model in a coupled climate model?

6,21: Formally speaking, SMB is not a boundary condition, but rather a source term in the mass conservation equation.

7,7: Suggest: "... that serve to evaluate the response of these initial states to ..."

7,10: "boundary conditions and assimilation targets" (plural)

7,17: "... analyzing models WITH RESPECT TO their individual ..."

8,8: "prescribe a fixed ice mask" – clarify what you mean here. I think you mean that they just zap away any ice moving past a certain location (e.g., present-day extent)?

Figure 1 caption: "A complete set of figures ... IS given ..."

9,1: "much smaller than to area." – provide a percentage value here?

9,1-3: I think the summary point here is that the areal expanse of the modeled ice sheets differs quite a bit w.r.t. observations, but that in terms of overall ice sheet volume, the differences are smaller (because even if you have an extensive error in the areal coverage, that marginal ice is presumably pretty thin).

9,10: Again, SMB is not really a boundary condition in any formal sense.

9,13: RCM = regional climate model? Not yet defined?

Figure 3: Note whether or not these 3 samples are from the middle of the distribution or if they span the distribution?

11,15: "good agreement with the observed" – awkward

11,15-16: ". . . is more important . . ." Not clear what "more important" is in reference to here. More important to what?

Figure 5: The units on the axes of figures in panels b and c are hard to read. Enlarge?

Figure 6 caption (and elsewhere): "Root mean square error (RMSE) . . ." (define "RMSE" used in figure axes labels).

13,9: "occurring at the margins" – be explicit here that what is important is that these errors are occurring over a fairly small fraction of the total ice sheet area?

13,17: "internal consistency" – this phrase is used fairly often without it being very clear what is meant here. I think in most places, it simply means that the long timescale ice sheet physics (like temperature) are closer to being in equilibrium with the relevant climate forcing (or have a reasonable and realistic transient).

14,1: ". . . at the expense of a larger discrepancy with the observed geometry." As noted above, more recent efforts include attempts to improve on this problem by including additional observational constraints beyond velocities (e.g., like the SMB forcing). See additional related discussion in Major Comments above.

**TCD**

14,9: "for almost any given rheology." – You may need to expand on this point for the non-initiated. What you mean is that you can generally get a good match to observed velocities by simply putting all of the motion into the sliding field and ignoring the fact that you might have the internal rheology entirely wrong.

Figure 7: The "v" and "s" symbols here are very hard to see.

15,2: "Recent MODELED mass trends . . ." ?

15,9: "which is currently not available" – Realistically though, if the data was available, would most of the models discussed here be able to apply it in a useful way? It's not really just the lack of data availability that is the problem here but also that the models aren't really in the position yet to make good use of these data.

15,10-13: And, the models trying to do hindcasting experiments generally still suffer from the same initialization problems you are focusing on here (either getting the initial ice sheet state right, or the trends right, but not both).

15,20-22: ". . . but more often . . . during initialization (i.e., the trends are dominated by the model's relaxation following an initialization "shock")."

16,30: Close with something like: "Simply put, by design a larger ice sheet will be subject to larger rates of mass loss." (note that one could try to correct for this by normalizing rates of mass loss relative to initial ice sheet volumes).

19,8: suggest: ". . . in other words, the models largely agree in their representation of the ice dynamical response to the applied SMB-anomaly forcing."

19,10-11: I don't know if I would call the range of approaches "wide". A more accurate description might be "representative".

19,20: "at the expense of long-term continuity." I don't follow what this means.

19,35: "differences in model ice density". I don't think this was discussed anywhere in the paper up until now.

---

## Referee Comment (RC2) · Anonymous Referee #2 · 18 Aug 2017

Review of Goelzer et al. "Design and results of the ice sheet model initialisation experiments initMIP-Greenland: an ISMIP6 intercomparison":

This manuscript describes the efforts of the ISMIP6 project to summarize various methods for initializing models of the Greenland Ice Sheet (GrIS) in preparation for experiments or projections of change. The impact of initialization is important as evidenced by previous ice sheet model inter-comparison (MIP) efforts, whose results were clouded by the 'uncontrolled-for' use of various initialization techniques. One impact of this is that initial GrIS conditions for projections across these earlier efforts varied widely, from conditions that (1) were constrained by the data assimilation and inversion techniques

to look almost like reality to (2) initial conditions that resulted from long freely-evolving equilibration simulations. This spread had a potentially significant impact on subsequent projections of sea level change.

Because of this issue I place value in this manuscript, which steps back to try and clearly illuminate the types of initialization techniques and their impact on perturbation experiments. I think as a survey of these techniques, this manuscript will be interesting to ice sheet modelers - including those who contributed to this MIP. In the following, my comments (which ignore minor grammar/editorial issues for now) are intended to (hopefully) improve the MIP by deepening analyses and interpretation of inter-model differences in the context of difference model choices (the main point of MIPs). In the few cases where I single out individual model results, be aware that I'm not criticizing these models/results per se, but only trying to use them to motivate a more general analysis. Finally I welcome any counter-arguments, if the authors feel I'm incorrect in any comments I've made.

Comments:

P3L12:"Changes in ice sheet geometry generally cause changes in atmospheric conditions over the ice sheet, and hence changes in SMB." Suggest that the word "feedback" be used here.

P3L13: "the most important effect is the height-SMB feedback" -> "an important effect is the height-SMB feedback" (other feedbacks aren't necessarily well-enough explored to make the 'most important' conclusion, in my opinion)

P3L17: for non-experts, perhaps also note for context the climatological fraction of total ice flux that comes from surface melt versus marine loss (instead of the fraction of current mass loss \*trends\* that arises from these two terms)

P3L27: "This is a very short period..."": Suggest adding some interpretation as to why these divergent timescales make things difficult.

P3L32: These two subsequent paragraphs (short-term vs. long-term projects) describe an issue that is similar to the difference between weather forecasting (an "initial value problem") and climate prediction (a "boundary value problem"). The difficult intermediate timeframe in the weather/climate context (seasonal->decadal prediction) is similar to the intermediate decadal->centennial timeframe in the ice sheet context. In both intermediate timeframes, it becomes unclear which technique (or blend of techniques) to use. I suggest describing this analogy to place the problem facing ice sheet modelers in the context of similar problems from other fields.

P3L32: suggest reordering this section to more clearly separate discussion of 'long-term' versus 'short-term' simulations and the initialization techniques used for each. As it stands the discussion mixes long and short time scale simulations a bit too much.

P4L15: "limitations in observations" -> "uncertainties in observations"?

P4L17: "transferred to" -> "masked by alterations to. . ." ..?

P4L24: Suggest a brief statement "Given the wide diversity of ice sheet initialization techniques. . ."

P4L24: "The goal of initMIP-Greenland is to document, compare and improve the techniques used by different groups to initialise their state-of-the-art whole-ice-sheet models to the present day. . .". This implies initialization to a transient ice sheet state (given anthropogenic forcing). For example, it appears that Goelzer et al. (2012) find a $\sim$0.7m GrIS SLR commitment if CO2 concentrations are capped at present day levels. Other studies I am aware of also find similar levels of SLR commitment to present-day climate. This arises from both historical warming relative to the preindustrial Holocene (which is the climate state most consistent with the current GrIS geometry) and also future committed warming, in very roughly equal parts. I think a discussion of this point is critical in order to put the stated goal, of initialization of ice sheets under present-day forcing/assimilation, in context. As one thought experiment: if the above two papers are correct in their commitment estimates, a perfect spin-up to equilibrium under perpetual present-day forcing should probably produce a present-day GrIS that is ~15% smaller than that observed. Another: a perfect DA procedure should result in an initial ice sheet with a transient trend equivalent to the recent historical average trend. Thus: would it be safer to remove "to the present day" from the statement to retain generality with respect to the chosen 'time of initialization'? Or state that initialization to present day is a choice driven largely by availability of observational data, which is required in the case of DA/DAv initialization techniques?

Table 3: Some rows are identical in their entries. For example, DMI-PISM1 and DMI-PISM2. The difference in these types of rows should be identified, so the reader can hopefully identify why they produce different results.

P9L5: I think these differences in approaches are very important points. It would be useful to be able to cross-list the initialized states (e.g. in Figure 2) against the amount of 'constraining' that the modeling groups used to obtain that state.

Figure comment: several colors in scatterplot figures (e.g. Figure 2) are very similar. For example, ARC-PISM and UAF-PISM3. Even on a good computer screen it is very difficult/impossible to tell the difference. Is there any way to label each dot (or replace each dot with a letter or number, perhaps also associated with a unique color)?

General presentation comment related to above 2x points: allowing users to interactively explore the MIP results would be incredibly useful, if The Cryosphere would allow links to supplementary interactive public online plots (perhaps hosted to/linked via the ISMIP6 website). As just one example: for each online plot, being able to isolate the the DA/DAv/SP contributor 'pools' and allowing an interactive data tip to identify model names upon hover-over, would be fantastic. One reasonably easy way of doing this is Plotly, I'm sure there are others. https://plot.ly/python/line-and-scatter/, "Line and Scatter Plots" example.

Suggestion: would the contributors consider providing a mask of grid points on the *terrestrial margins* of their ice sheets that are (unphysical, at steady state) accumulation or (physically consistent) ablation zones? In my experience, it is possible for the terrestrial margin points from RCMs to potentially display positive SMB (presumably due to RCM bias). If a significant number of contributors to this MIP are using SMB fields that include these types of unphysical terrestrial margin accumulation zones, an important discussion could be: how these are handled and how the presence of this unphysical margin behavior could affect future simulations.

P9L15: "Again, a good match with the observed ice extent is more important than the SMB model itself.": this statement, as a value judgement, is unclear to me.

P13L14: "However, it is noticeable that DA models that have been initialised with one data set show lower errors when comparing with that specific set of observations.": this seems a reflection of a well-known statistical tenet, that one can't use training data as validation data. Perhaps this similarity should be more obviously stated for readers. Put another way: to what extent is comparing against observations useful, when in many cases (DA/DAv) these observations may been already used as part of the initialization procedure?

Figure 4 (and general question): if an ISM is allowed to grow outside of observed boundaries, but receives SMB from an RCM that is limited in extent to the present-day ice ice extent, how is the SMB calculated outside of the present-day ice extent?

Figure 6: Models for which SMB is prescribed to be strongly negative outside the current ice extent will also show low RMSE thickness. I'm aware of at least one participating model that takes this approach. I would be surprised if other well-performing spin-up models did not do the same. Models that take such an approach should be identified so that their good RMSE thickness performance can be judged against the strong constraint of very negative SMBs outside of current ice extent (which will likely, by basic ice dynamics principles, translate into a pretty good ice sheet geometry).

General comment: it is difficult to assess the an unbiased spread of results when a few models are represented by multiple contributions. I'm not sure how one would remedy

this, but it leaves the reader a little confused - especially when the difference between some contributions from the same model is not immediately clear (see previous comment).

P16L16: "...relatively small mass change for most of them over the course of the 100-year experiment...": when contrasted against observed/projected sea level rise rates, I suspect many of these rates of change, even after 100 years, are very large (for example, relative to realistic future GrIS SLR rates). One way to better show ctrl rates of change would be to show the anomaly of each model, relative to the initial volume (which is already represented anyways in Figure 2).

P16L19: "...when nine obvious outliers are ignored": it seems that the role of a MIP would be to explain why these outliers exist instead of discarding them. For example, did these models use DA/DAv/SP?

P16L21: "In some cases of the ensemble, the modelled background trend arises from transient forcing of SMB and temperature over the past, but more often it is due to inconsistencies introduced during initialisation." Is it possible to identify which are which (potentially via online plotting technique mentioned above)?

Figure8b: at what point in the simulations is the 'mass trend in control' (y-axis) assessed?

P17L21: "This relation arises because the prescribed SMB anomaly has been optimized for the observed geometry, but has not been limited to the observed ice sheet extent." As I understand, this means that the prescribed SMB anomaly gets larger the farther one goes from the present-day margin (Appendix A). Thus, it seems the strong relationship between initial size (and closely related area) and sea level response is mainly a direct response to a design feature of the MIP, and is not an emergent property of the ISMs themselves. Because of this I question the initial volume/SLR response relationship the authors highlight.

P18L5: I appreciate that a real SLR projection would probably use the 'perturbation minus control' approach to remove drift. However, for the sake of a MIP, I feel that showing the raw results would be very useful, so readers can truly appreciate what each model is actually doing. Conversely, perhaps the authors could make a plot/table showing the ratio of drift to perturbation magnitudes for each model, so readers could better assess how significant the drift actually is (instead of assessing this via visual plot comparisons). For example, this would allow users to assess their favorite rule of thumb for acceptable drift (for example, mine is: drift magnitude must be 1/10 of the expected signal magnitude).

General: I think the supplementary figures may be of most interest to the participants in this MIP. I really appreciate their inclusion.

Figure S3: for models like AWI and ISSM whose elevation is not defined outside of the present-day mask, what happens to ice that flows out of the present day mask? Is it simply set to zero..?

Figure S3: suggest outlining the ice sheet margin in red, or another obvious color, so bare land vs. ice sheet can be distinguished.

Figure S4: It is clear from this figure which models allow their ice sheets to evolve outside of the present-day volume, and which do not. As above, I'd suggest noting explicitly in a table which models do so and which don't.

Figure S5 (subsetted in Figure 10): A reader will immediately question why the ctrl thickness change for many DA* models (AWI-ISSM2,AWI-ISSM1,IGE-ELMER1, IGE-ELMER2, UCIJPL-ISSM, MIROC-ICIES1, ULB-FETISH2) have such major dH/dt artifacts, and why these artifacts won't artificially impact SLR projections (by control-minus-perturbation or absolute change methods). In the MIP spirit, I'd like the underlying reason for this pattern described, and in close conjunction, also why other DAv-based models do NOT exhibit this pattern and why. I think there may be important lessons here. I'd suggest a section of text discussing this.
Figure S5 (subsetted in Figure 10): Similar to the above comment: several SP models display large ctrl trends (e.g. PISM1-6). This is surprising since the point of a spin-up is generally to remove such trends. Conversely, other SP models have much less of a trend. Along the lines of the previous comment, I'd suggest a section of text discussing why differences in ctrl trends for SP models arise. There could be additional important lessons here.

Figures S6/S7: The negative ends of these color ranges indicates (I think) that the SMB forcing is dominant over the dynamic response in these experiments (which exclude potential ocean-driven dynamical forcing). First of all, isolating the dynamic response is a nice addition here. Secondly, this is slightly interesting with respect to the overall MIP, since it indicates that the applied SMB anomaly is a very important player here (especially with respect to initial ice sheet size); ice dynamics are essentially second order. Perhaps if the authors agree with me they could add a discussion to this effect (or provide a counterargument).

P20L20: I think this is a valuable summary paragraph. Suggest again making an analogy to weather forecasts versus climate projections (and/or initial value versus boundary value problems).

P20L33: "potential artefacts introduced during interpolation": can the authors possibly provide a quantitative estimate of this source of error?

P20L37: "The large ensemble spread in sea-level contribution in the asmb experiment is mostly due to the extra ice in the initial ice sheet geometry." Yes, but as above I'd argue that this relationship is mostly a consequence of the design of the imposed SMB anomaly field. As the authors note, a more realistic anomaly field that takes into account ice sheet geometry bias (e.g. a lapse-rated PDD scheme or an EBM scheme on multiple elevation classes) would likely show a much less pronounced volume-response relationship. So as stated, it's not clear this is a problem to focus community efforts on exclusively (though, of course the less initial volume bias the bet-

ter, as long as the bias reduction technique doesn't deleteriously impact subsequent projections).

General: thanks to all the participants/coordinators of this MIP. This was no small effort.

---

## Author Comment (AC1) · 9 Oct 2017

**We have revised our manuscript 'Design and results of the ice sheet model initialisation experiments initMIP-Greenland: an ISMIP6 intercomparison'. We would like to thank the reviewers for their constructive comments that helped to improve the manuscript.**

**Please find below the reviewer's comments in regular italic and a point-by-point response in bold font.**

*Anonymous Referee #1*

*Summary*
*This paper summarizes models, methods, and the results of ice sheet model initialization experiments for the Greenland ice sheet, which are (eventually) targeting the ISMIP6 contribution to CMIP6. The broad motivation for the exercise is the recognition – largely motivated by previous, community-endorsed ice sheet model intercomparisons – that decadal and century scale simulations conducted using present-day ice sheet models are keenly sensitive to the model initial conditions (so much so that transients in the initial condition often swamp the ice sheet model response to the prescribed climate forcing of interest). In this paper, the different initialization methods most commonly used by the ice sheet modeling community are discussed, as are the perceived pros and cons of the different approaches. A set of forward model experiments are conducted, which are intended to clearly expose the magnitude and trend of (1) model initialization transients under steady forcing and (2) model transients under idealized climate forcing, here applied in the form of surface mass balance (SMB) anomalies.*

*While the results of the simulations and paper are not necessarily in themselves scientifically compelling, they serve as an important record and waypoint for the purpose of documenting the current practices and capabilities of different modeling groups, as well as the differences in standard model outputs when applying these different models to the same set of experiments. Therefor, while this paper does not necessarily represent any major scientific breakthroughs, I think it is worthy of publication in TC purely for the purpose of clearly documenting where the ice sheet modeling community stands on these issues at this time. The paper represents the combined and significant efforts of a large number of independent ice sheet modeling groups, including the countless hours spent on model development and testing prior to applying the models to any experiments.*
*I disagree a bit with the way some of the conclusions for the experiments have been presented or "pitched" here. For one, it's not clear to me that the spread and/or size of the transients in the steady-forcing initialization experiment do actual show improvement over previous efforts. This could be due, in part, to the way the data are presented*

*(discussed further below). It could also be that a simple quantitative metric from this study, relative to previous studies, needs to be included to support this statement. Further, the fact that the perturbation experiment uses anomalies applied on top of the SMB used for initialization is already an indication that we aren't looking at an apples- to-apples comparison of response between models. For example, if an actual SMB forcing (as opposed to a set of anomalies) was prescribed for the perturbation experiment, the spread in sea-level-equivalent mass loss shown in Fig. 8c would likely be much larger. I don't necessarily disagree with this choice, but the summary statements don't seem entirely accurate.*

*One obvious place for improvement is in the conclusions, which currently do not provide any clear guidance as to a "path forward", in the sense of what did we learn here and where should the community be working towards in order to address some of the issues and problems discussed in this paper. It seems like one obvious conclusion that could be stated more clearly is that either using data-assimilation OR spin-up approaches in isolation is not a long-term solution. Rather, using some combination of these two approaches is going to be necessary if the goal is to initialize models to have both a realistic initial state while simultaneously minimizing unrealistic transients (good) or representing actual observed ice sheet transients (better). There are some references (noted below) that could be included for these types of approaches, which aren't currently being applied at the whole-ice-sheet scale, but which should be viable for doing so in the near future. This would help to end the paper on a more positive, forward-looking note.*

*Below, comments are tied to specific points in the submitted text using the key "X, Y", where X refers to the page number and Y refers to the line number.*

*Major Comments*

*1,18: I'm not sure I really agree with the last sentence. If the authors really think this is true, then they should back it up quantitatively somewhere in the text. Is the drift referred to here for the init. and steady-forcing experiment? The SMB anomaly experiment? Both?*

**OK, the sentence has been reformulated. We refer to the drift in the control experiment, where models that can be compared (because they have also participated in the SeaRISE experiments) all show less drift.**

*2,3: "encapsulates most of the modeling decisions . . ." – I don't entirely agree with this. A major "modeling decision" would be whether or not I choose to use SIA or a Stokes model, but I can imagine using the same init. procedure for both if I chose to do so. It might make more sense here to just point out that the decisions made during the init. process have a very strong impact on any model outputs that follow from using that initial state. I think the point you are trying to make here is simply that these choices are critical as they have very large consequences for "projections".*

**Yes, agreed, but we also wanted to emphasise that many aspects of the modelling are revealed when looking at initialisation. To follow on the given example, one wouldn't really initialise with the SIA version of a model and then run that state forward with the Stokes version. In other words, many of the modelling decisions have to be made before initialisation. Replaced "encapsulates" by "therefore reveals".**

*2,28-30: The last sentence here is not really true. For example, there are fairly extensive "internal" constraints available in the form of radar layers, dated at ice core sites and tracked over hundreds of km of the ice sheet. Even though we are not regularly using these data now as constraints, we should be, or could be in the future. Maybe this is a point to make in the conclusions / future directions part of the paper, with appropriate references to some of the work by Joe MacGregor (which is particularly relevant to Greenland)?*

**OK. We have added a sentence to discuss this additional information in the text. A discussion point on further use of the radar layer data has been added to the conclusions.**

*3,28-30: "All three methods . . .". Note that these could / can all be combined, and this is probably the future direction we as a community should be moving towards. It might be worth noting that here, even though none of the currently participating models do this (and then discuss this again in the conclusions by pointing to a few papers that are exploring what would truly be called data assimilation – that is, trying to assimilate observations of change (like dh/dt) and/or fields that include a record of change over time (like temperature, layer shapes)).*

**OK. We have added a point in the conclusions to discuss the combination of methods and assimilation approaches.**

*3,7-33: In general, I think the major problem being discussed here could be stated a bit more succinctly. That is, the spin-up approach provides an initial condition that has a realistic internal TRANSIENT with respect to some applied external climate forcing, but also generally gives us an initial STATE that is a poor representation of the present day. The current incarnation of optimization approaches does the opposite: we get a good representation of the present-day ice sheet state, but a very bad representation of the internal transients. The solution going forward should be to formally combine these two seemingly at-odds approaches through improved data assimilation approaches.*

**We discuss this problem in detail in the first two paragraphs of section 5 and have added additional material as an outlook what future research should aim for.**

*5,2-3: It is probably important to be explicit about the "lineage" of models here. While there are 17 group contributions, more than half of these are from "repeated" models (i.e., 9 of the models listed are either ISSM, PISM, or SICOPOLIS), so in effect, there are only really 11 different models represented here.*

**OK. Added "There is some overlap between the code bases used by different groups, with ultimately 11 individual ice flow models. However, the same model used by different groups (with varying datasets and initialisation procedures) may lead to very different results".**

*6,2-23: It might be nice to point out that the division of the three initialization strategies is a bit arbitrary, and not perfect. For example, most of the models that use data assimilation also use some form of spin-up (e.g., for ice temperatures). Also, the "DAs" method is really a sort of "ad hoc" approach to assimilation – instead of using formal methods to iteratively minimize a PDE-constrained, observation-minus-model cost functional, these approaches are minimizing the observation-minus-model difference using some other method (i.e., adjusting the basal traction locally to minimize the mismatch with surface elevations). I appreciate that some categorization needs to be made here to bin these approaches, but it's worth pointing out that the categorization is far from perfect. It's actually quite fuzzy and this is because it's starting to be obvious that some combination of these different methods is required.*

**Agreed. We have added additional material to clarify that the separation is somewhat arbitrary and that combinations exist and should be further explored.**

*10,5-21: It's not entirely clear to me what the purpose of this section is. It's missing a clearly motivating intro / closing sentence to make that clear.*

**The purpose of this section is to evaluate the SMB forcing used by different models and includes figures 3-5. This intent is given in the paragraph just before:**
**"As an important input for the ice sheet simulations, we evaluate the implemented SMB of the different models"**

*Figure 8: If you are going to keep panel (a) of this figure as is, then I think you also need to show a 2nd figure where you normalize by the initial volume (in SLE). As shown, these trends all look very reasonable but this is because you are showing the rate of absolute, rather than relative change. When normalized by the initial volumes, we would more easily see the volume changes with the more relevant units of cm rather than m. In panels (b) and (d), what time periods are the trends calculated over? In panel (c), the vertical axis label is "mass change". For this, a positive sign would suggest ice sheet growth. Should the wording be changed (e.g., "mass loss") so that it is clear we are looking at mass loss (which is obvious if we are talking about SLR)?*

**The purpose of panel (a) is primarily to display the range of initial ice mass (vertical scale), but also to give an impression of the overall mass evolution. Since all mass curves in (a) are near constant or monotonically increasing or monotonically decreasing, the information in panel (b) gives a very good idea of the magnitude of mass change (vertical scale in mm!). Normalising by the volume would be a different way of looking at the data that we don't favour for this comparison. E.g. models with too large initial mass and large mass change would in relative terms tend to look 'better' than they are.**

**We have changed the labels in panels (a,b,c) as suggested and also the sign of the values in b, to match a. Note, the period of mass change (100 years) is already given in the caption.**

*16,9: "inconsistencies between ice velocity and geometry datasets". The reference to Seroussi et al. won't help the unfamiliar reader here. I suggest you just be explicit about what the problem is: the optimized model velocities, while possibly being consistent with observed surface velocities, conspire with thickness errors to give a flux divergence that is wildly different than and unbalanced by the local SMB. This is where the large and*

*unrealistic thickness transients come from when conducting a forward run following initialization using a velocity-based-only optimization approach. Note that the uppermost panel in Figure 10 shows this nicely – the very noisy, large amplitude thickness adjustments in the ISSM model are a manifestation of this (relative to the much more subtle thickness transients for the spun-up ICIES2 and PISM models).*

**OK. We have included a statement as suggested to name the problem explicitly and added to the discussion of figure 10.**

*19,16-18: This is left hanging a bit. As noted above in the summary comments, instead of simply pointing out that none of the methods discussed here prove to be optimal for initialization, you could take the opportunity to discuss recent efforts that target these deficiencies (even if they are not yet recognizable at the full-ice sheet scale) and push for their exploration, development, and wider adoption. I'm thinking in particular about formal efforts that attempt to include additional constraints and / or assimilate time dependent observations (for example, Goldberg et al., TC, 7(6), 2013; Perego et al., J. Geophys. Res., 119, 2014).*

**OK. We have added to the discussion of these points and references, and suggest further exploration of these approaches**

*19,20: "DAv is the method of choice for short-term projections." I strongly disagree with this statement. Unless you are talking about applying anomaly forcing (as in the perturbation experiment discussed here), this is absolutely not true. Most often, these are going to have the worst and most non-physical thickness transients when stepped forward in time under realistic climate forcing. Only if that climate forcing has been taken into consideration during the optimization process do these have a hope of minimizing non-physical transients. This can clearly be seen in the top left figure of Figure 10 (the ISSM, DAv initialized model has a noisy, large amplitude thickness transient that is very unphysical). In general, this paragraph is not true unless you are talking about anomaly forcing experiments, which I think in general, the community is trying to move away from (for example, an initial condition generated to work well only when applied in an anomaly forcing experiment is of little use for coupling to a climate model, since climate models are generally required to conserve mass, and anomaly forcing does not do that).*

**Agreed. Reformulated to explicitly limit to cases of anomaly forcing, which is of interest here. This is how we have forced the models and what will very likely be**

**used for the ISMIP6-CMIP projections, because it is not feasible to initialise to a large number of different GCMs.**

*20,7-9: I am not yet convinced that this statement – here and in the abstract – is supported by the results shown in this paper. It could be the case, but based on Figure 8a, it is very hard to judge. We would have to see Figure 8a normalized by initial ice sheet volume to be convinced of this. Also, if this statement is going to be supported, it would be nice to do so quantitatively based on previous intercomparison results (e.g. compare the spread from a similar experiment from SeaRISE with that from this study).*

**OK. We have reformulated this statement. Models that can be compared (because they have also participated in the SeaRISE experiments) all show less drift.**

*20,11-13: "If this trend continues . . .". While I agree with this statement, I don't think it is for the same reasons as discussed in this paper. If there's any reason this trend will continue to improve, it's that better formal optimization and initialization methods are being developed and applied to ice sheets. That is, methods that take into account not only observational constraints on the current ice sheet state, but also observational constraint on short (i.e. dh/dt) and longer (i.e., temperature profiles) term trends that are inherent to the ice sheet. Again, I feel like there is a bit of a missed opportunity in the conclusions of this paper to point out and promote some of these new directions and methods that should effectively minimize many of the problems highlighted and discussed in this paper.*

**OK. We have added to the discussion pointing out further directions of research.**

*Minor / Editorial Comments*

*Title: Check that the spelling of "initialisation" is correct? Autocorrect seems to prefer "initialization".*

**In BE both appear to be possible. We prefer "initialisation" in line with the other British spellings.**

*1,2-3: Should summary refs. for the SeaRISE and Ice2Sea projects be included here? Similarly, should ISMIP6 and CMIP6 efforts here point to some generic reference publications?*

**Good point, but no references allowed in the abstract. Included in introduction instead.**

*1,6: ". . . to estimate the associated uncertainties." (in what? Model outputs?)*

**OK. Added "in modelled mass changes".**

*1,10: schematic -> idealized ?*

**OK. Also all other occurrences.**

*1,13-14: Is there really a wide diversity in the data sets and boundary conditions used? It seems like for the most part, the models are using similar datasets and boundary conditions (for the latter, I mean w.r.t. "observation-based" boundary conditions – I understand that things like basal boundary conditions, which include assumptions about sliding, thermal state, hydrology, etc., probably DO vary widely).*

**OK. Removed "wide". The intention is to make clear that the diversity that exists in the community is represented here.**

*1,21-28: The initial reference to EISMINT, followed by a listing of many other non-EISMINT intercomparisons, is confusing. I understand that you only mean to use the categorization that EISMINT suggested, but it could be read as if everything underneath (e.g., ISMIP-HOM, MISMIP, etc.) was also "part of" EISMINT. Suggest rewording this slightly?*

**OK. Reformulated to "EISMINT and later following comparisons include".**

*1, general: Do CMIP and ISMIP6 need to be defined here?*

**Thanks, definitions are included now.**

*2,10: "removing mass at the margins" – this is a little unclear, as SMB also removes mass inland (although maybe this mass is transferred to the margins, e.g. as melt).*

**OK. Added "predominantly" to clarify this point.**

*2,32-34: I would be more emphatic about this point – without very special care, the transient from initialization can entirely dominate the response in decadal / century scale projections.*

**The wording "are strongly influenced" seems appropriate to us.**

*2,41: When you say run forward in time here, it might help to add detail about the timescale you are thinking about (e.g., thousands to tens of thousands (more?) years, rather than decades or centuries).*

**OK. Added "for tens to hundreds of thousands of years".**

*3, 2: "the model's state is internally consistent" – It isn't clear what you mean by this. The model's internal state is always internally consistent, regardless of the type of spin-up you do. I think what you mean is that models that go through a spin-up are, at any point in time, closer to being in equilibrium with the applied climate forcing, or at least contain internal transients (e.g., in temperature) that are more consistent with external forcing transients. I appreciate this is a subtle difference to try to express.*

**OK. Reformulated to "the model's state is defined as a consistent response to the forcing". We believe this is the decisive point, while equilibrium with the forcing is not required/ often not desired.**

*3,10: "by inversion" – be a bit more specific here. I think you mean formal, PDE-constrained optimization?*

**OK. Reformulated to "by a formal partial differential equations constrained optimisation".**

*3,16-18: Other "errors" that are forced to be accounted for by optimized parameter fields include observations and model state variables that are entirely ignored in the optimization process. For example, one of the biggest reasons current optimization*

*approaches lead to such large, unphysical transients is that they are ignoring whether or not the temperature (and hence rheology) is consistent with observations (and as such, all of that uncertainty gets pushed into the basal parameter optimization).*

**OK. Reformulated.**

*3,19-22: Here, I think you need to be explicit in noting that the thing that is being adjusted is the bedrock elevation (i.e., we have good constraints on the ice sheet surface elevation, but much less so with the bedrock elevations. So the thickness is adjusted with the assumption that the uncertainty is the bedrock location, not the upper surface location).*

**The method discussed here (and in Pollard and DeConto 2012) doesn't involve modifying the bedrock elevation. The models participating in initMIP-Greenland only adjust for the basal friction coefficient in their data assimilation of surface elevation (see also the model descriptions in the appendix).**

*4,5: "to a large perturbation" – add "in SMB forcing"? or "in climate forcing"?*

**OK. added "in SMB forcing"        .**

*4,17: ". . . that all boundary conditions AND FORCING remain constant in time." ?*

**OK.**

*Table 1: "schematic change of SMB forcing" -> "idealized change in SMB forcing"*

**OK.**

*Table 2: Is the list of "contributors" consistent with co-authors on the paper, or is it supposed to be more of a list of co-authors for the respective modeling efforts?*

**The latter. There is a lot of overlap, but not all contributors are necessarily co-authors on the paper.**

*6,12-14: Note that SP approaches are generally also favored when the goal is to include an ice sheet model in a coupled climate model?*

**OK. Included "SP approaches are also generally favoured when including ice sheets in coupled climate models.", before "In two groups …".**

*6,21: Formally speaking, SMB is not a boundary condition, but rather a source term in the mass conservation equation.*

**OK. Removed "boundary condition".**

*7,7: Suggest: ". . . that serve to evaluate the response of these initial states to ..."*

**Thanks. Reformulated as suggested.**

*7,10: "boundary conditions and assimilation targets" (plural)*

**OK.**

*7,17: ". . . analyzing models WITH RESPECT TO their individual . . ."*

**OK.**

*8,8: "prescribe a fixed ice mask" – clarify what you mean here. I think you mean that they just zap away any ice moving past a certain location (e.g., present-day extent)?*

**OK. Added "and prevent any ice growing outside".**

*Figure 1 caption: "A complete set of figures . . . IS given . . ."*

**OK.**

*9,1: "much smaller than to area." – provide a percentage value here*

**The volume difference between the two defined areas is 0.3 % while differences in area are 8.2 %. These numbers have been included in the manuscript.**

*9,1-3: I think the summary point here is that the areal expanse of the modeled ice sheets differs quite a bit w.r.t. observations, but that in terms of overall ice sheet volume, the*

*differences are smaller (because even if you have an extensive error in the areal coverage, that marginal ice is presumably pretty thin).*

**This is true for the observed (see response to last point) but not for the modelled results (15% max error in area and volume alike). Models that overestimate area typically exhibit thicker margins than observed, accommodating additional volume.**

*9,10: Again, SMB is not really a boundary condition in any formal sense.*

**OK. Replaced "boundary condition" by "input".**

*9,13: RCM = regional climate model? Not yet defined?*

**This has already been defined at 7,21.**

*Figure 3: Note whether or not these 3 samples are from the middle of the distribution or if they span the distribution?*

**OK. Added "spanning the distribution of ice thickness error (see below)"**

*11,15: "good agreement with the observed" – awkward*

**OK. Added "SMB"**

*11,15-16: ". . . is more important . . ." Not clear what "more important" is in reference to here. More important to what?*

**OK. Added**
**"to reduce the mismatch with measured SMB"**

*Figure 5: The units on the axes of figures in panels b and c are hard to read. Enlarge?*

**OK. The labels have been enlarged. Same for figure 7.**

*Figure 6 caption (and elsewhere): "Root mean square error (RMSE) ..." (define "RMSE" used in figure axes labels).*

**OK. Modified here and in three other places.**

*13,9: "occurring at the margins" – be explicit here that what is important is that these errors are occurring over a fairly small fraction of the total ice sheet area?*

**OK. Reformulated to**
**"occurring at the margins over a relatively small fraction of the ice sheet area are weighted less"**

*13,17: "internal consistency" – this phrase is used fairly often without it being very clear what is meant here. I think in most places, it simply means that the long timescale ice sheet physics (like temperature) are closer to being in equilibrium with the relevant climate forcing (or have a reasonable and realistic transient).*

**What was meant by "internal consistency" was really that the dynamics of the system are a consistent response to the forcing and not a response to the initialisation. We have removed "internal" in all cases and clarified what is consistent with what in all cases.**

*14,1: ". . . at the expense of a larger discrepancy with the observed geometry." As noted above, more recent efforts include attempts to improve on this problem by including additional observational constraints beyond velocities (e.g., like the SMB forcing). See additional related discussion in Major Comments above.*

**This is discussed with additional references included in section 5.**

*14,9: "for almost any given rheology." – You may need to expand on this point for the non-initiated. What you mean is that you can generally get a good match to observed velocities by simply putting all of the motion into the sliding field and ignoring the fact that you might have the internal rheology entirely wrong.*

**OK. Added**
**", as all the uncertainty (including unknown rheology) is compounded in the basal sliding relation."**

*Figure 7: The "v" and "s" symbols here are very hard to see.*

**OK. We have increased the font size of the symbols.**

*15,2: "Recent MODELED mass trends . . ." ?*

**Yes. Included 'modelled'.**

*15,9: "which is currently not available" – Realistically though, if the data was available, would most of the models discussed here be able to apply it in a useful way? It's not really just the lack of data availability that is the problem here but also that the models aren't really in the position yet to make good use of these data.*

**We agree. That is why our original statement implies not only missing data but also missing process understanding and implementation ("realistic outlet glacier dynamics and ocean forcing"). Not changed.**

*15,10-13: And, the models trying to do hindcasting experiments generally still suffer from the same initialization problems you are focusing on here (either getting the initial ice sheet state right, or the trends right, but not both).*

**OK. Reformulated to**
**"suffer the same limitations of observational data sets with short time coverage, uncertainty in external forcing, limited knowledge of processes responsible for dynamic outlet glacier response and the initialisation problems discussed here."**

*15,20-22: ". . . but more often . . . during initialization (i.e., the trends are dominated by the model's relaxation following an initialization "shock").''*

**OK. Included, but reformulated to "(i.e., the trends are dominated by the model's response to the initialisation, not to the forcing)". We prefer not to use the term "shock".**

*16,30: Close with something like: "Simply put, by design a larger ice sheet will be subject to larger rates of mass loss." (note that one could try to correct for this by normalizing rates of mass loss relative to initial ice sheet volumes).*

**Sentence has been added as suggested.**

**We are not in favour of normalizing to initial ice sheet volume. It arguably reduces the model spread, but for the wrong reasons.**

*19[20],8: suggest: ". . . in other words, the models largely agree in their representation of the ice dynamical response to the applied SMB-anomaly forcing."*

**OK.**

*19[20],10-11: I don't know if I would call the range of approaches "wide". A more accurate description might be "representative".*

**OK.**

*19,20: "at the expense of long-term continuity." I don't follow what this means.*

**OK. Reformulated to**
**"at the expense of including long-term processes", which should become clear in context of the sentence before.**

*19,35: "differences in model ice density". I don't think this was discussed anywhere in the paper up until now.*

**We have added some additional information on how SLE is calculated (including differences in density) in the description of figure 8, where it already occurred once in caption.**

*Anonymous Referee #2*

*Review of Goelzer et al. "Design and results of the ice sheet model initialisation experiments initMIP-Greenland: an ISMIP6 intercomparison":*
*This manuscript describes the efforts of the ISMIP6 project to summarize various methods for initializing models of the Greenland Ice Sheet (GrIS) in preparation for experiments or projections of change. The impact of initialization is important as evidenced by previous ice sheet model inter-comparison (MIP) efforts, whose results were clouded by the 'uncontrolled-for' use of various initialization techniques. One*

*impact of this is that initial GrIS conditions for projections across these earlier efforts varied widely, from conditions that (1) were constrained by the data assimilation and inversion techniques to look almost like reality to (2) initial conditions that resulted from long freely-evolving equilibration simulations. This spread had a potentially significant impact on subsequent projections of sea level change.*

*Because of this issue I place value in this manuscript, which steps back to try and clearly illuminate the types of initialization techniques and their impact on perturbation experiments. I think as a survey of these techniques, this manuscript will be interesting to ice sheet modelers - including those who contributed to this MIP. In the following, my comments (which ignore minor grammar/editorial issues for now) are intended to (hopefully) improve the MIP by deepening analyses and interpretation of inter-model differences in the context of difference model choices (the main point of MIPs). In the few cases where I single out individual model results, be aware that I'm not criticizing these models/results per se, but only trying to use them to motivate a more general analysis. Finally I welcome any counter-arguments, if the authors feel I'm incorrect in any comments I've made.*

*Comments:*

*P3L12:"Changes in ice sheet geometry generally cause changes in atmospheric conditions over the ice sheet, and hence changes in SMB." Suggest that the word "feedback" be used here.*

**Yes, we use 'feedback' in the sentence directly following P3L12.**

*P3L13: "the most important effect is the height-SMB feedback" -> "an important effect is the height-SMB feedback" (other feedbacks aren't necessarily well-enough explored to make the 'most important' conclusion, in my opinion)*

**OK, reformulated as suggested.**

*P3L17: for non-experts, perhaps also note for context the climatological fraction of total ice flux that comes from surface melt versus marine loss (instead of the fraction of current mass loss \*trends\* that arises from these two terms)*

**Thanks for the clarification. The balance of processes is indeed what we meant to describe. The sentence has been reformulated.**

*P3L27: "This is a very short period. . .": Suggest adding some interpretation as to why these divergent timescales make things difficult.*

**OK, added a sentence to clarify that problem.**

*P3L32: These two subsequent paragraphs (short-term vs. long-term projects) describe an issue that is similar to the difference between weather forecasting (an "initial value problem") and climate prediction (a "boundary value problem"). The difficult intermediate timeframe in the weather/climate context (seasonal->decadal prediction) is similar to the intermediate decadal->centennial timeframe in the ice sheet context. In both intermediate timeframes, it becomes unclear which technique (or blend of techniques) to use. I suggest describing this analogy to place the problem facing ice sheet modelers in the context of similar problems from other fields.*

**OK, added a sentence to elude to this analogy.**

*P3L32: suggest reordering this section to more clearly separate discussion of 'long- term' versus 'short-term' simulations and the initialization techniques used for each. As it stands the discussion mixes long and short time scale simulations a bit too much.*

**We are not really mixing time scales here. The first paragraph P3L32 describes what we want to do: project evolution starting at the present to the (100 year) future. Then follows a description of the two/three predominant approaches used to initialise to the present day, which have different typical time scales attached. We have tried to make that clearer by replacing "used" by "developed" in "Models developed for long-term and paleoclimate simulations".**

*P4L15: "limitations in observations" -> "uncertainties in observations"?*

**OK. Reformulated as suggested.**

*P4L17: "transferred to" -> "masked by alterations to. . ." ..?*

**Not changed. Important to say that errors are propagated to the optimised parameters.**

*P4L24: Suggest a brief statement "Given the wide diversity of ice sheet initialization techniques. . ."*

**Thanks. Included as suggested.**

*P4L24: "The goal of initMIP-Greenland is to document, compare and improve the techniques used by different groups to initialise their state-of-the-art whole-ice-sheet models to the present day. . .". This implies initialization to a transient ice sheet state (given anthropogenic forcing). For example, it appears that Goelzer et al. (2012) find a ~0.7m GrIS SLR commitment if CO2 concentrations are capped at present day levels. Other studies I am aware of also find similar levels of SLR commitment to present-day climate. This arises from both historical warming relative to the preindustrial Holocene (which is the climate state most consistent with the current GrIS geometry) and also future committed warming, in very roughly equal parts. I think a discussion of this point is critical in order to put the stated goal, of initialization of ice sheets under present-day forcing/assimilation, in context. As one thought experiment: if the above two papers are correct in their commitment estimates, a perfect spin-up to equilibrium under perpetual present-day forcing should probably produce a present-day GrIS that is ~15% smaller than that observed. Another: a perfect DA procedure should result in an initial ice sheet with a transient trend equivalent to the recent historical average trend. Thus: would it be safer to remove "to the present day" from the statement to retain generality with respect to the chosen 'time of initialization'? Or state that initialization to present day is a choice driven largely by availability of observational data, which is required in the case of DA/DAv initialization techniques?*

**This question has been discussed further below in the manuscript in the section on the transient experiments (P16). We have made clear that matching the observed imbalance is not really possible in the present framework. The period we refer to as 'present-day' is necessarily a bit loosely defined because we wanted to give freedom to the modellers what year to initialise to.**

*Table 3: Some rows are identical in their entries. For example, DMI-PISM1 and DMI-PISM2. The difference in these types of rows should be identified, so the reader can hopefully identify why they produce different results.*

**OK. Have included further details for DMI-PISM, and for other models that have identical rows (UAF-PISM 1-3 and 4-6).**

*P9L5: I think these differences in approaches are very important points. It would be useful to be able to cross-list the initialized states (e.g. in Figure 2) against the amount of 'constraining' that the modeling groups used to obtain that state.*

**We agree in principle with the idea of providing this information. However, we don't see how the amount of 'constraining' should be quantified in a meaningful way. This is a problem we face with such a diverse ensemble, which we discuss at different places in the manuscript.**

*Figure comment: several colors in scatterplot figures (e.g. Figure 2) are very similar. For example, ARC-PISM and UAF-PISM3. Even on a good computer screen it is very difficult/impossible to tell the difference. Is there any way to label each dot (or replace each dot with a letter or number, perhaps also associated with a unique color)?*

**The presentation of the data is always a compromise between showing the ensemble as a whole and showing individual models. Our choice is deliberate and emphasises the ensemble, while still making it possible to trace back information on an individual model if desired. The data in Fig 2, for example, is available in the supplement for readers who want to know the details.**

*General presentation comment related to above 2x points: allowing users to interactively explore the MIP results would be incredibly useful, if The Cryosphere would allow links to supplementary interactive public online plots (perhaps hosted to/linked via the ISMIP6 website). As just one example: for each online plot, being able to isolate the the DA/DAv/SP contributor 'pools' and allowing an interactive data tip to identify model names upon hover-over, would be fantastic. One reasonably easy way of doing this is Plotly, I'm sure there are others. https://plot.ly/python/line-and-scatter/, "Line and Scatter Plots" example.*

**Thanks for the suggestion, which we are currently exploring. As stated in the manuscript, the data will be made publically available and users will be able to display the data as desired.**

*Suggestion: would the contributors consider providing a mask of grid points on the \*terrestrial margins\* of their ice sheets that are (unphysical, at steady state) accumulation or (physically consistent) ablation zones? In my experience, it is possible*

*for the terrestrial margin points from RCMs to potentially display positive SMB (presumably due to RCM bias). If a significant number of contributors to this MIP are using SMB fields that include these types of unphysical terrestrial margin accumulation zones, an important discussion could be: how these are handled and how the presence of this unphysical margin behavior could affect future simulations.*

**It is not entirely clear to us what the reviewer's suggestion is aiming for and how that is related to the manuscript. We interpret the question as largely evolving around the issue of how to use climate model output to force an ice sheet model. From the perspective of this paper, this is an issue that needs to be resolved by individual modellers. In some cases, climate model biases are much larger and need different strategies than in other cases. This question will become very important for ISMIP6 when we design the ice sheet projections, but arguably has been largely circumvented in the present anomaly forcing approach.**

*P9L15: "Again, a good match with the observed ice extent is more important than the SMB model itself.": this statement, as a value judgement, is unclear to me.*

**See similar point by reviewer 1. We have added "to reduce the mismatch with measured SMB" to clarify the intention of this sentence,**

*P13L14: "However, it is noticeable that DA models that have been initialised with one data set show lower errors when comparing with that specific set of observations.": this seems a reflection of a well-known statistical tenet, that one can't use training data as validation data. Perhaps this similarity should be more obviously stated for readers. Put another way: to what extent is comparing against observations useful, when in many cases (DA/DAv) these observations may been already used as part of the initialization procedure?*

**We believe there are two sides to this issue that are both represented in the manuscript. The first is the point about training and validation data made at P13L8 and P14L9. However, to evaluate the quality of an assimilation process, one should in fact look at how well it can reproduce the training data set.**

*Figure 4 (and general question): if an ISM is allowed to grow outside of observed boundaries, but receives SMB from an RCM that is limited in extent to the present-day ice ice extent, how is the SMB calculated outside of the present-day ice extent?*

**This is a well-known problem, but the solution is left to the individual modellers. Typical approaches consist of parameterising the SMB or its components as a function of surface elevation.**

*Figure 6: Models for which SMB is prescribed to be strongly negative outside the current ice extent will also show low RMSE thickness. I'm aware of at least one participating model that takes this approach. I would be surprised if other well-performing spin-up models did not do the same. Models that take such an approach should be identified so that their good RMSE thickness performance can be judged against the strong constraint of very negative SMBs outside of current ice extent (which will likely, by basic ice dynamics principles, translate into a pretty good ice sheet geometry).*

**We are definitely aware of this and related questions of 'fairness' for model evaluation and caution the reader in this regard (see e.g. again P13L8). However, a negative SMB is also a very strong condition in the real world for ice not being present in certain places. The model descriptions in Appendix B generally include this information, but approaches are too diverse to allow for a binary flag.**

*General comment: it is difficult to assess the an unbiased spread of results when a few models are represented by multiple contributions. I'm not sure how one would remedy this, but it leaves the reader a little confused - especially when the difference between some contributions from the same model is not immediately clear (see previous comment).*

**Being well aware of this problem, we never attempt to present the results as unbiased. Furthermore, the problem runs deeper, because some now separate model families have common ancestors that would need to be revealed.**

*P16L16: ". . .relatively small mass change for most of them over the course of the 100-year experiment. . .": when contrasted against observed/projected sea level rise rates, I suspect many of these rates of change, even after 100 years, are very large (for example, relative to realistic future GrIS SLR rates). One way to better show ctrl rates of change would be to show the anomaly of each model, relative to the initial volume (which is already represented anyways in Figure 2).*

**The corresponding numbers are shown in Fig 8 panel b and given as range in the text (~-20 to +25 mm when outliers are removed). We wanted to represent the spread of initial mass in panel a and have complemented that with mass change in panel b. Please also see comment to reviewer 1.**

*P16L19: ". . .when nine obvious outliers are ignored": it seems that the role of a MIP would be to explain why these outliers exist instead of discarding them. For example, did these models use DA/DAv/SP?*

**These models (except for one) have already been discussed as outliers in context of figure 4, so this was not repeated here. However, we have added "discussed in context of Figure 4" to remind the reader.**

*P16L21: "In some cases of the ensemble, the modelled background trend arises from transient forcing of SMB and temperature over the past, but more often it is due to inconsistencies introduced during initialisation." Is it possible to identify which are which (potentially via online plotting technique mentioned above)?*

**We have reformulated to "In some cases of the ensemble (typically for the SP models)" to identify the models where this should typically be the case. However, any change in the model setup from initialisation to control run can generate a drift and cannot be exclude for any of the models.**

*Figure8b: at what point in the simulations is the 'mass trend in control' (y-axis) assessed?*

**Over the 100-year simulation period as described in the caption.**

*P17L21: "This relation arises because the prescribed SMB anomaly has been optimized for the observed geometry, but has not been limited to the observed ice sheet extent." As I understand, this means that the prescribed SMB anomaly gets larger the farther one goes from the present-day margin (Appendix A). Thus, it seems the strong relationship between initial size (and closely related area) and sea level response is mainly a direct response to a design feature of the MIP, and is not an emergent property of the ISMs themselves. Because of this I question the initial volume/SLR response relationship the authors highlight.*

**To clarify, the SMB anomaly (as a function of elevation) gets mostly smaller (i.e. more negative) towards the coast of Greenland. This aside, the discussion in this mentioned paragraph indeed clarifies that the relationship seen in Fig8d results to some extent from the experimental design. However, ice sheets with too large ice extent also show large volume losses because they are in balance with an unrealistic SMB. It would be possible to define an experimental design where differences due to different ice extent would be smaller (e.g. anomaly adopted to individual geometries) but we disagree that this would be better/more realistic.**

*P18L5: I appreciate that a real SLR projection would probably use the 'perturbation minus control' approach to remove drift. However, for the sake of a MIP, I feel that showing the raw results would be very useful, so readers can truly appreciate what each model is actually doing. Conversely, perhaps the authors could make a plot/table showing the ratio of drift to perturbation magnitudes for each model, so readers could better assess how significant the drift actually is (instead of assessing this via visual plot comparisons). For example, this would allow users to assess their favorite rule of thumb for acceptable drift (for example, mine is: drift magnitude must be 1/10 of the expected signal magnitude).*

**We believe the combination of information from the different panels in figure 8 gives a good overview of the ensemble. Tabled results are available in the supplement for the interested reader.**

*General: I think the supplementary figures may be of most interest to the participants in this MIP. I really appreciate their inclusion.*

**Thank you very much.**

*Figure S3: for models like AWI and ISSM whose elevation is not defined outside of the present-day mask, what happens to ice that flows out of the present day mask? Is it simply set to zero..?*

**For these models, ice leaves the ice sheet mask only at calving locations, like in the other models. Note that the caption states "where surface elevation is not defined outside of the ice sheet mask", which is typically different from the observed present-day mask.**

*Figure S3: suggest out lining the ice sheet margin in red, or another obvious color, so bare land vs. ice sheet can be distinguished.*

**OK. We have added contour lines to delineate the ice sheet mask.**

*Figure S4: It is clear from this figure which models allow their ice sheets to evolve outside of the present-day volume, and which do not. As above, I'd suggest noting explicitly in a table which models do so and which don't.*

**Please see response to comment above. Note that ice volume may be constrained during initialisation in some of the models, but never in the forward runs. Area is effectively controlled in some cases, but again, that is difficult to quantify. See also response to comment on figure 6.**

*Figure S5 (subsetted in Figure 10): A reader will immediately question why the ctrl thickness change for many DA\* models (AWI-ISSM2,AWI-ISSM1,IGE-ELMER1, IGE-ELMER2, UCIJPL-ISSM, MIROC-ICIES1, ULB-FETISH2) have such major dH/dt artifacts, and why these artifacts won't artificially impact SLR projections (by control-minus-perturbation or absolute change methods). In the MIP spirit, I'd like the underlying reason for this pattern described, and in close conjunction, also why other DAv-based models do NOT exhibit this pattern and why. I think there may be important lessons here. I'd suggest a section of text discussing this.*

**We agree that this aspect has not received enough attention in the manuscript before. We have therefore added a further discussion of figure 10, which is summarising results for the different models as far as possible.**

*Figure S5 (subsetted in Figure 10): Similar to the above comment: several SP models display large ctrl trends (e.g. PISM1-6). This is surprising since the point of a spin-up is generally to remove such trends. Conversely, other SP models have much less of a trend. Along the lines of the previous comment, I'd suggest a section of text discussing why differences in ctrl trends for SP models arise. There could be additional important lessons here.*

**It is not correct that SP methods attempt to remove trends. Rather, the trends should arise physically consistent from (past) changes in the forcing. Please see**

**response to the question before. We now discuss the response in experiment ctrl in more detail.**

*Figures S6/S7: The negative ends of these color ranges indicates (I think) that the SMB forcing is dominant over the dynamic response in these experiments (which exclude potential ocean-driven dynamical forcing). First of all, isolating the dynamic response is a nice addition here. Secondly, this is slightly interesting with respect to the overall MIP, since it indicates that the applied SMB anomaly is a very important player here (especially with respect to initial ice sheet size); ice dynamics are essentially second order. Perhaps if the authors agree with me they could add a discussion to this effect (or provide a counterargument).*

**Indeed, our results show that the dynamic response to SMB forcing alone (ignoring other forcing mechanism like ocean-induced changes) is a relatively small effect. However, that does not represent a new finding, it was already demonstrated by Huybrechts et al., 2002, which we discussed in the main manuscript and therefore, we do not see the need to re-iterate this here.**

*P20L20: I think this is a valuable summary paragraph. Suggest again making an analogy to weather forecasts versus climate projections (and/or initial value versus boundary value problems).*

**Thanks for the suggestion, we believe mentioning this once in the introduction is sufficient.**

*P20L33: "potential artefacts introduced during interpolation": can the authors possibly provide a quantitative estimate of this source of error?*

**We have not been able to quantify that error, but think it is nevertheless important to mention it as a possible source of differences between models. We have added a statement to clarify that.**

*P20L37: "The large ensemble spread in sea-level contribution in the asmb experiment is mostly due to the extra ice in the initial ice sheet geometry." Yes, but as above I'd argue that this relationship is mostly a consequence of the design of the imposed SMB anomaly field. As the authors note, a more realistic anomaly field that takes into account ice sheet geometry bias (e.g. a lapse-rated PDD scheme or an EBM scheme on multiple elevation*

*classes) would likely show a much less pronounced volume-response relationship. So as stated, it's not clear this is a problem to focus community efforts on exclusively (though, of course the less initial volume bias the better, as long as the bias reduction technique doesn't deleteriously impact subsequent projections).*

**The larger ensemble spread is really a combination of models producing too large initial ice sheets and an experimental design that emphasizes this. We disagree that an SMB anomaly "that takes into account ice sheet geometry bias" would be more** *realistic***. One could argue that it would effectively represent a work-around for model problems to simulate an ice sheet close to the observed geometry. See also response to comment above.**

*General: thanks to all the participants/coordinators of this MIP. This was no small effort.*

**Thanks again for reviewing this paper.**